# Distinct functions of POT1 proteins contribute to the regulation of telomerase recruitment to telomeres

Peili Gu[1], Shuting Jia[2], Taylor Takasugi [1], Valerie M. Tesmer[3], Jayakrishnan Nandakumar [3], Yong Chen [4] & Sandy Chang [1,5,6✉]

Human shelterin components POT1 and TPP1 form a stable heterodimer that protects telomere ends from ATR-dependent DNA damage responses and regulates telomerase-dependent telomere extension. Mice possess two functionally distinct POT1 proteins. POT1a represses ATR/CHK1 DNA damage responses and the alternative non-homologous end-joining DNA repair pathway while POT1b regulates C-strand resection and recruits the CTC1-STN1-TEN1 (CST) complex to telomeres to mediate C-strand fill-in synthesis. Whether POT1a and POT1b are involved in regulating the length of the telomeric G-strand is unclear. Here we demonstrate that POT1b, independent of its CST function, enhances recruitment of telomerase to telomeres through three amino acids in its TPP1 interacting C-terminus. POT1b thus coordinates the synthesis of both telomeric G- and C-strands. In contrast, POT1a negatively regulates telomere length by inhibiting telomerase recruitment to telomeres. The identification of unique amino acids between POT1a and POT1b helps us understand mechanistically how human POT1 switches between end protective functions and promoting telomerase recruitment.

[1] Department of Laboratory Medicine, Yale University School of Medicine, New Haven, CT 06520, USA. [2] Laboratory of Molecular Genetics of Aging and Tumor, School of Medicine, Kunming University Science and Technology, Kunming 650500, China. [3] Department of Molecular, Cellular, and Developmental Biology, University of Michigan, Ann Arbor, MI 48109, USA. [4] State Key Laboratory of Molecular Biology, Shanghai Institute of Biochemistry and Cell Biology, Center for Excellence in Molecular Cell Science, Chinese Academy of Sciences, Shanghai 200031, China. [5] Department of Pathology, Yale University School of Medicine, New Haven, CT 06520, USA. [6] Department of Molecular Biophysics and Biochemistry, Yale University School of Medicine, New Haven, CT 06520, USA. ✉email: schang@yale.edu

Telomeres are TTAGGG repetitive DNA-ribonucleoprotein complexes that cap the ends of all eukaryotic chromosomes[1,2]. Telomeres terminate in G-rich single-stranded (ss) DNA overhangs and thus have the potential to initiate inappropriate DNA repair reactions. However, ss G-overhangs are also required for telomere elongation by telomerase. A major question in the telomere field is how telomeres are able to protect chromosome ends from being recognized as damaged DNA while still functioning as substrates for telomerase-mediated telomere elongation. A potential solution to this problem resides in a complex of telomere-specific binding proteins, termed shelterin, which protects telomeres from inappropriately activating DNA damage checkpoints[1,3]. Three sequence-specific DNA-binding proteins are recruited to

**Fig. 1 POT1b promotes telomere elongation. a** PNA-telomere FISH shows telomeric signals present in G3 $Pot1b^{+/-}$ and G3 $Pot1b^{-/-}$ sarcoma cells. TelG: PNA probe Cy3-OO-(CCCTAA)$_4$. Red arrows point to telomere-free chromosome fusions and white arrows point to telomere-free chromosome ends. Scale bar: 5 μm. **b** Distribution of telomere lengths, designated as arbitrary telomere intensity, in G3 $Pot1b^{+/-}$ and G3 $Pot1b^{-/-}$ sarcomas determined by quantitative (Q) telomere-FISH. Telomeres from a minimum of 30 metaphases were scored in three independent experiments. Numbers indicate median telomere lengths. **c** Quantification of the indicated types of fusions per chromosome in $Pot1b^{-/-}$ sarcomas of the indicated generations. Data show the mean ± standard deviation (s.d.) from three independent experiments. At least 30 metaphases were analyzed for each sample. **d** PNA-FISH of G3 $Pot1b^{-/-}$ sarcoma expressing GFP, POT1a$^{WT}$, POT1b$^{WT}$ or POT1b$^{F62A}$ constructs. TelG: PNA probe Cy3-OO-(CCCTAA)$_4$. Red arrows point to telomere-free chromosome fusions; white arrows point to telomere-free chromosome ends. Scale bar: 5 μm. **e** Distribution of telomere intensities by Q-FISH from metaphases in (**d**). A minimum of 40 metaphases were scored per cell type in three independent experiments. **f** Quantification of telomere-free ends in (**d**). Data show the mean ± s.d. from two independent experiments. At least 30 metaphases were analyzed per construct. p-values are shown and generated from one-way ANOVA analysis followed by Tukey's multiple comparison. **g** Quantification of telomere-free chromosome fusions in (**d**). Data show the mean ± s.d. from two independent experiments. At least 30 metaphases were analyzed per construct. p-values are shown and generated from one-way ANOVA analysis followed by Tukey's multiple comparison. **h** Detection of G-overhangs (native gel) and total telomere lengths (denature gel) by TRF Southern in G3 $Pot1b^{-/-}$ sarcomas expressing the indicated constructs. Ethidium bromide (EtBr) image shows that all samples contained equal amounts of genomic DNA. Molecular weight markers are indicted. *: DNA band used for quantification. Numbers indicate relative G-overhang and total telomere signals, with telomere signals set to 1.0 for cells expressing GFP.

chromosomal ends: the duplex telomere binding proteins TRF1 and TRF2 and the telomere ssDNA binding protein Protection of Telomeres 1 (POT1). POT1 forms a functional heterodimer with TPP1, and in turn, TPP1 tethers POT1 to telomeres by interacting with the TIN2-TRF1 and TIN2-TRF2 complexes[4]. POT1 homologs have been identified in most eukaryotes[5]. While most vertebrates, including humans, possess a single POT1 gene, multiple POT1 paralogs with distinct functions have been found in worms[6], plants[7,8] and mice. The mouse genome encodes two POT1 genes, POT1a and POT1b, due to a recent gene duplication[9–12]. POT1 proteins contain two highly conserved OB folds that interact with the 3′ terminus of the telomere ssDNA overhang to promote telomere end protection. Human POT1-TPP1 binds to telomeric DNA to enhance the processivity of telomerase in vitro[13,14]. While TPP1 directly recruits telomerase to telomere through its N-terminal TEL patch[15,16], conflicting biochemical and genetic evidence reveal that human POT1 functions both as a negative and a positive regulator of telomerase activity at telomeres[9,17,18].

We and others have shown that POT1a and POT1b possess distinct functions at telomeres. POT1a protects telomeres from being recognized as damaged DNA[11,12,19,20]. Many proteins involved in the DNA damage response, including phosphorylated γ-H2AX, 53BP1, MRN1, ATM/ATR, and CHK1/2, localize rapidly to telomeres devoid of functional POT1a. We have shown previously that deletion of POT1a in mice results in early embryonic lethality[12] and activation of an ATR-Chk1 DDR at telomeres, leading to inappropriate HDR-mediated chromosome fusions[12,21,22].

POT1b functions to maintain the 3′ G-overhang. Overhang formation in mice requires nucleolytic processing of the 5′ C-strand by exonucleases Apollo/SNM1B and Exo1[23,24]. Additionally, POT1b recruits the CTC1-STN1-TEN1 (CST) complex to modulate DNA Polymerase-α fill-in synthesis of the C-strand[25–27]. Consequently, deletion of POT1b in mice does not induce a potent DDR but increases G-overhang length while accelerating telomere shortening[17,28]. Mice lacking POT1b survive to adulthood but eventually succumb to bone marrow failure by ~14 months of age. Telomere length analysis revealed that telomeres in POT1b null hematopoietic stem cells (HSCs) are extremely short[17,29]. In addition, deletion of POT1b, coupled with telomerase haploinsufficiency, induces accelerated telomere shortening, culminating in rapid onset of bone mallow failure by ~6 months of age. These results suggest that in addition to its role in repressing C-strand resection, POT1b is required for telomere length maintenance in both somatic cells and HSCs.

While the molecular basis for these functional differences between POT1a and POT1b remains poorly understood, it is important to note that human POT1 exhibits the functions of both proteins. In this report, we show that POT1b plays a pivotal role in telomere elongation. POT1b directly interacts with TPP1 to promote telomerase recruitment to telomeres and concomitant telomere elongation. Our results suggest that in addition to its known role in recruiting CST-DNA Pol-α to promote C-strand fill-in, POT1b is also required to promote telomerase dependent G-strand synthesis.

## Results

**POT1b promotes telomere elongation.** To understand how the POT1a and POT1b proteins contribute to telomere maintenance, we bred $Pot1a^{F/F}$, $Pot1b^{-/-}$ and $p53^{F/F}$ mice to generate $Pot1a^{F/F}$; $Pot1b^{-/-}$; $p53^{F/F}$ and $Pot1a^{F/F}$; $Pot1b^{+/-}$; $p53^{F/F}$ mouse embryonic fibroblasts (MEFs). Deletion of both $Pot1a$ and $Pot1b$ alleles resulted in early embryonic lethality so we were unable to obtain $Pot1a/b$ double null MEFs. We treated MEFs with Adeno-CMV-Cre and after selection in culture, obtained immortalized $Pot1a^{F/\Delta}$; $Pot1b^{-/-}$; $p53^{\Delta/\Delta}$ and $Pot1a^{F/\Delta}$; $Pot1b^{+/-}$; $p53^{\Delta/\Delta}$ MEFs. Having one floxed $Pot1a$ allele protected telomere ends, enabling cells to proliferate normally and abolished the generation of dysfunctional telomere induced DNA damage foci (TIFs). These cells also readily formed tumors when injected into the flanks of severe combined immunodeficient (SCID) mice (Supplementary Fig. 1a, b). After sarcoma formation, we harvested the tumors to produce first-generation (G1) sarcoma cell lines. We injected $1 \times 10^5$ G1 $Pot1a^{F/\Delta}$; $Pot1b^{-/-}$; $p53^{\Delta/\Delta}$ sarcoma cells into SCID mice to generate G2 sarcomas and $1 \times 10^3$ G2 cells into SCIDs to generate G3 sarcomas (Supplementary Fig. 1a).

Compared to G3 $Pot1a^{F/\Delta}$; $Pot1b^{+/-}$; $p53^{\Delta/\Delta}$ cells (abbreviated as G3 $Pot1b^{+/-}$), quantitative telomere (Q)-FISH analysis of metaphase spreads revealed that G3 $Pot1a^{F/\Delta}$; $Pot1b^{-/-}$; $p53^{\Delta/\Delta}$ cells (abbreviated as G3 $Pot1b^{-/-}$) possess extremely short telomeres (mean arbitrary telomere intensity of ~2000 vs. ~6000, respectively) and a dramatic reduction in the number of chromosome ends possessing telomere signal intensities greater than 5000 mean arbitrary telomere intensity (Fig. 1a, b; Supplementary Fig. 1c, d). In addition, examination of parental, G1, G2 and G3 sarcomas reveals a progressive increase in the number of end-to-end chromosome fusions lacking any telomere signals and a corresponding decrease in the number of fusions bearing telomere signals at fusion sites (Fig. 1c, Supplementary Fig. 1e, f). We reasoned that the very short telomeres observed in G3 $Pot1b^{-/-}$ sarcomas should make these cells highly sensitive to

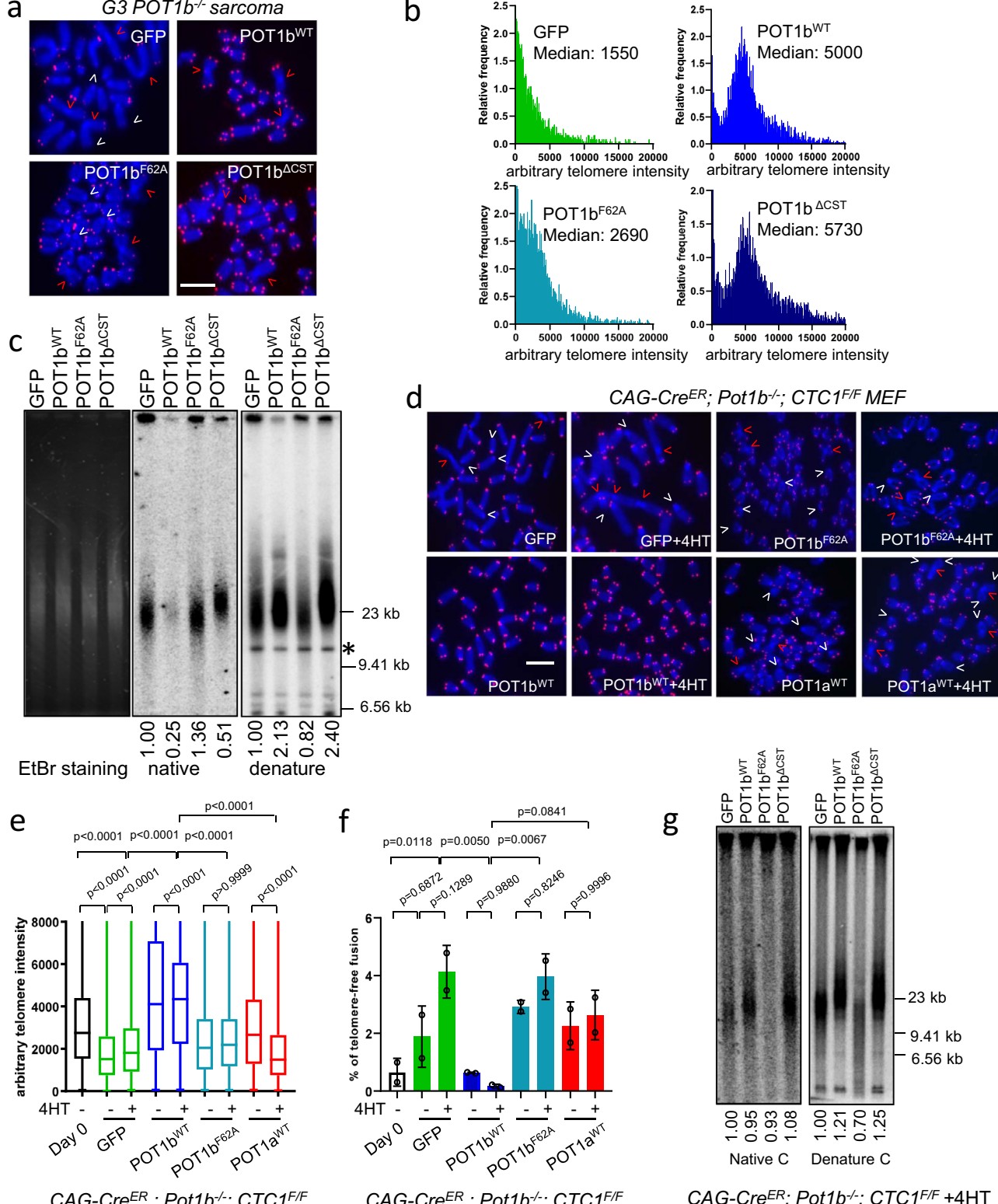

examine the impacts of POT1a and POT1b on telomere length maintenance. Intriguingly, Q-FISH analysis revealed that expressing POT1b^WT, but not GFP control, in G3 *Pot1b^{-/-}* sarcomas for 30 days resulted in rapid telomere elongation, with ~2-fold increase in both median telomere length and the number of chromosome ends now bearing longer telomeres (Fig. 1d, e; Supplementary Fig. 1g–i). Analysis of 40 metaphase spreads from

POT1b^WT-reconstituted G3 *Pot1b^{-/-}* sarcomas revealed that 100% of metaphases examined now display a ~7-fold decrease in the number of telomere-free chromosome ends, suggesting elongation of the shortest telomeres after POT1b^WT expression (Fig. 1f, Supplementary Fig. 1g–i). Not surprisingly, the level of background end-to-end chromosome fusions was unaffected, since activation of telomere elongation pathways is not expected

**Fig. 2 POT1b-mediated telomere elongation is independent of CST function. a** PNA-FISH of G3 *Pot1b*$^{-/-}$ sarcomas reconstituted with the indicated DNA constructs. Red arrows point to telomere-free chromosome fusions; white arrows point to telomere-free chromosome ends. Scale bar: 5 μm. **b** Distribution of telomere lengths by Q-FISH in (**a**). A minimum of 30 metaphases were analyzed in three independent experiments. Numbers indicate median telomere lengths. **c** Detection of G-overhangs (native gel) and total telomere lengths (denature gel) by TRF Southern. *: DNA band used for quantification. EtBr: Ethidium Bromide. Numbers indicate relative G-overhang and total telomere signals, with telomere signals set to 1.0 for cells expressing GFP. **d** PNA-FISH of *CAG-Cre*$^{ER}$; *Pot1b*$^{-/-}$; *CTC1*$^{F/F}$ MEFs ± 4-HT reconstituted with indicated POT1a or POT1b. Red arrows point to telomere-free chromosome fusions and white arrows point to telomere-free chromosome ends. Scale bar: 5 μm. **e** Quantification of telomere lengths in (**d**) by Q-FISH. At least 30 metaphases were analyzed per genotype. Box show the median and interquartile range (25% to 75%) from three independent experiments. p-values are shown and generated from one-way ANOVA analysis followed by Tukey's multiple comparison. **f** Quantification of telomere-free chromosome fusions in (**d**). At least 30 metaphases were analyzed per genotype. Data show the mean ± s.d. from two independent experiments. p-values are shown and generated from one-way ANOVA analysis followed by Tukey's multiple comparison. **g** TRF Southern to detect the G-overhang (native gel) and total telomere lengths (denature gel) of *CAG-Cre*$^{ER}$; *Pot1b*$^{-/-}$; *CTC1*$^{F/F}$ MEFs + 4-HT, expressing the indicated DNAs. Molecular weight markers are indicated. Numbers indicate relative G-overhang and total telomere signals, with telomere signals set to 1.0 for cells expressing GFP.

to impact chromosome ends that have already fused (Fig. 1g). Quantitative pulse-field telomere restriction fragment (TRF) Southern analysis revealed that expression of POT1b$^{WT}$ increased the total telomere length of G3 *Pot1b*$^{-/-}$ sarcomas by ~2-fold over cells expressing GFP-vector control, with a corresponding decrease in the amount of single-stranded G-overhang observed (Fig. 1h).

To understand mechanistically how POT1b promotes telomere elongation, we asked whether expression of POT1a$^{WT}$ or the POT1b$^{F62A}$ mutant unable to interact with single-stranded telomeric 3′ G-overhangs[12] is able to elongate telomeres when expressed in G3 *Pot1b*$^{-/-}$ sarcomas. Expression of similar levels of POT1a$^{WT}$ or POT1b$^{F62A}$ as POT1b$^{WT}$ (Supplementary Fig. 1g, h) was unable to induce telomere lengthening, indicating that POT1a cannot substitute for POT1b's function in telomere elongation. POT1b$^{F62A}$ expression was unable to lead to telomere extension, reinforcing our observation that interaction of the POT1b N-terminal OB-folds with the 3′ G-overhang is essential for POT1b-mediated telomere extension (Fig. 1e, f, h; Supplementary Fig. 1i).

To exclude the possibility of cell-type-dependent telomere lengthening by POT1b, we reconstituted POT1b$^{WT}$ into several independently derived early generation *Pot1b*$^{-/-}$ MEFs bearing longer telomeres. Similar to what we observed in G3 *Pot1b*$^{-/-}$ sarcomas, only reconstituting POT1b$^{WT}$, but not POT1a$^{WT}$ nor POT1b$^{F62A}$, led to ~2-fold increase in mean arbitrary telomere intensity (Supplementary Fig. 2a, b). Both Q-FISH and TRF Southern revealed that POT1b$^{WT}$ expression also extended telomere lengths in SV40-immortalized G2 *Pot1b*$^{-/-}$ sarcomas and in a *Pot1b*$^{-/-}$; *p53*$^{P/P}$ mutant MEF cell line bearing very short telomeres[30] (Supplementary Fig. 2c–e). These observations suggest that POT1b, but not POT1a, is able to promote telomere length elongation in *Pot1b*$^{-/-}$ sarcomas and MEFs.

**The CST complex and Rad51 are not required for telomere length elongation by POT1b.** POT1b directly interacts with CTC1 and recruits the CST complex to telomeres to promote C-strand fill-in by DNA polymerase alpha[27,31]. The disappearance of the telomeric 3′ G-overhang when POT1b$^{WT}$ was expressed in G3 *Pot1b*$^{-/-}$ sarcomas (Fig. 1h) suggests that reconstituted POT1b$^{WT}$ recruits CST to fill-in the telomeric C-strand and generate blunt-ended telomeres. We postulated that the recruitment of the CST-Pol-α to telomeres might be responsible for the telomere elongation phenotype observed after POT1b$^{WT}$ expression. To test this hypothesis, we utilized the POT1b$^{ΔCST}$ mutant unable to interact with CTC1 to recruit the CST complex to telomeres[27]. Q-FISH and TRF Southern analyses of G3 *Pot1b*$^{-/-}$ sarcomas reconstituted with either POT1b$^{ΔCST}$ or POT1b$^{WT}$ revealed telomere lengths in cells expressing POT1b$^{ΔCST}$ to be slightly longer than those expressing POT1b$^{WT}$

(Fig. 2a–c, Supplementary Fig. 3a). This result suggests that POT1b's recruitment of the CST complex to telomeres is not required for telomere length elongation. Our data also supports a role for the CST complex in repressing murine telomere elongation, in agreement with its role as a terminator of human telomere elongation[25]. However, the observed reduction of the 3′ G-overhang after POT1b$^{WT}$ reconstitution is completely dependent on POT1b$^{WT}$'s ability to recruit the CST complex to telomeres to promote C-strand fill-in (Figs. 1h, 2c).

To further confirm that the CST complex is not required for POT1b-mediated telomere elongation, we generated *CAG-Cre*$^{ER}$; *Pot1b*$^{-/-}$; *CTC1*$^{F/F}$ MEFs which enabled 4-hydroxytamoxifen (4-HT)-mediated deletion of the floxed *CTC1* alleles[26]. Q-FISH analysis revealed that only reconstitution of POT1b$^{WT}$, but not POT1b$^{F62A}$ or POT1a$^{WT}$, into 4-HT treated *CAG-Cre*$^{ER}$; *Pot1b*$^{-/-}$; *CTC1*$^{Δ/Δ}$ MEFs resulted in a significant increase in total telomere length (Fig. 2d, e, g; Supplementary Fig. 3b–d). POT1b$^{WT}$ expression also significantly reduced the number of chromosome fusions lacking telomere signals at fusion sites (telomere-free fusions), suggesting that telomeres elongated by the addition of POT1b$^{WT}$ were no longer dysfunctional and therefore do not participate in further end-to-end fusion reactions (Fig. 2d, f). In contrast, expression of GFP control, POT1b$^{F62A}$ or POT1a$^{WT}$ failed to increase telomere length and did not reduce the number of ongoing chromosome fusions (Fig. 2e–g, Supplementary Fig. 3d). Since the expression of POT1b$^{WT}$ and POT1b$^{ΔCST}$ elongated both the G-overhang and total telomere length in the absence of endogenous CTC1, our results indicate that POT1b$^{WT}$-mediated telomere length extension is independent of its ability to recruit the CST complex to telomeres.

Telomere elongation could also occur through the engagement of a Rad51-dependent Alternative Lengthening of Telomeres (ALT) pathway[32]. To determine whether G3 *Pot1b*$^{-/-}$ sarcomas utilize ALT to elongate telomeres, we used shRad51 to efficiently deplete Rad51 and then reconstituted either GFP vector or POT1b$^{WT}$ (Supplementary Fig. 4a–c). Depletion of Rad51 did not negatively impact the telomere length elongation induced by POT1b$^{WT}$ expression, suggesting that POT1b$^{WT}$-mediated telomere lengthening does not depend upon the activation of a Rad51-dependent homologous recombination pathway (Supplementary Fig. 4d, e).

**Telomere elongation by POT1b is dependent on telomerase.** We next examined whether POT1b plays a role in enhancing telomerase's ability to elongate telomeres. To address this question, we used CRISPR/CAS9 to delete the *mTert* gene in G3 *Pot1b*$^{-/-}$ sarcomas. Out of 192 colonies screened, we were able to obtain two clones (#27 and #44) in which one *mTert* allele was rendered nonfunctional with a frameshift mutation and the other *mTert* allele bearing an in-frame 11 amino acid deletion in the reverse transcriptase domain (Fig. 3a and Supplementary Fig. 5a, b). TRAP

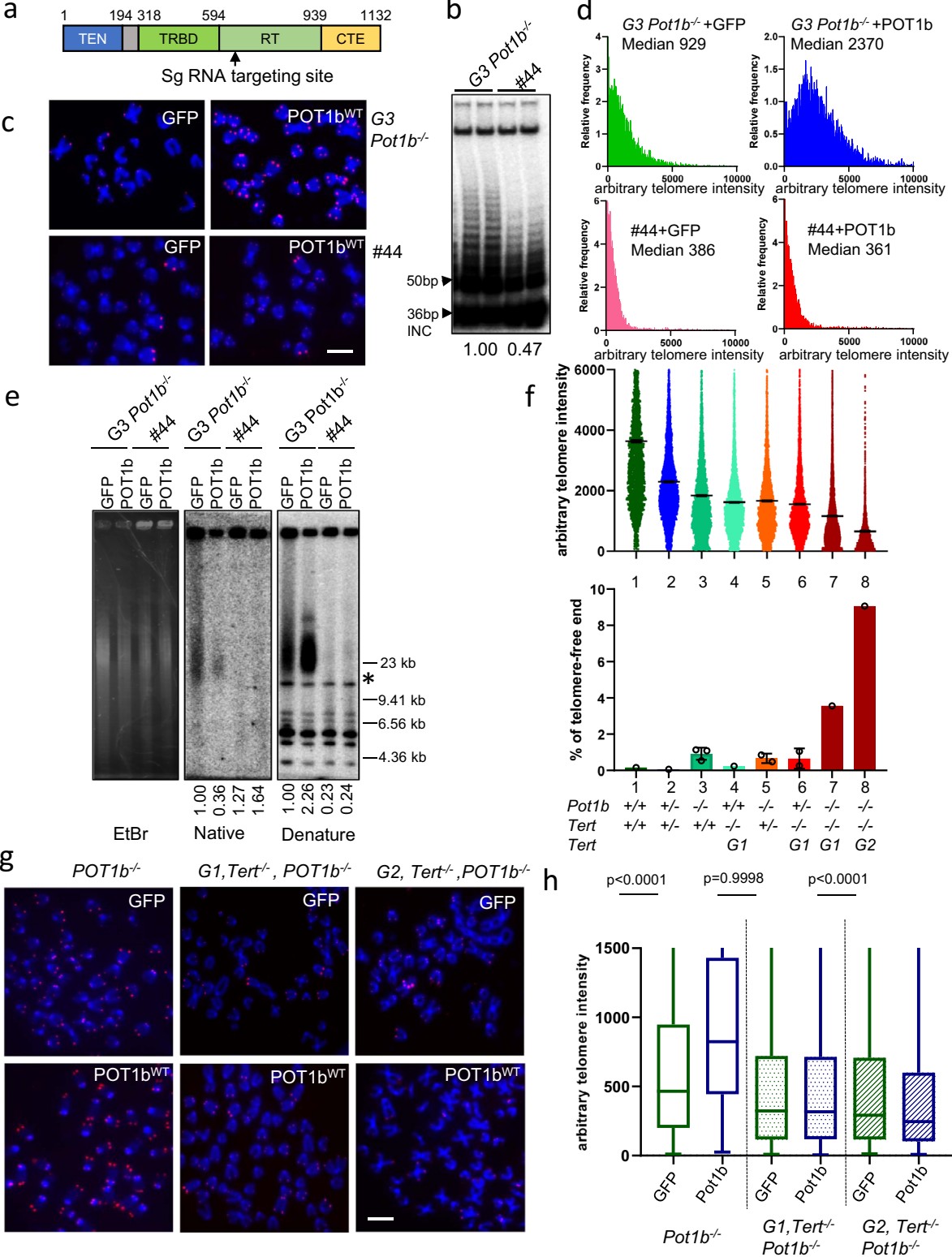

assays revealed that clone #44 is hypomorphic for telomerase activity, displaying only ~47% of WT telomerase levels (Fig. 3b). Q-FISH and TRF Southern analysis reveal that compared to G3 *Pot1b*−/− cells, both TERT hypomorphic clones displayed ~2-3-fold shorter telomeres that did not elongate after reconstitution with POT1b[WT] (Fig. 3c–e and Supplementary Fig. 5c–e). In contrast, the G3 *Pot1b*−/− clone #32 that failed to undergo *mTert* gene editing exhibited telomere elongation after expressing POT1b[WT]. These

results suggest that the telomere elongation mediated by POT1b[WT] in G3 *Pot1b*−/− sarcomas requires wild-type telomerase levels.

To confirm this hypothesis, we crossed *mTert*+/− mice with *Pot1b*+/− mice to generate G1 and G2 *mTert*−/−; *Pot1b*−/− MEFs. Compared to *Pot1b*−/− MEFs, telomere PNA-FISH analysis of G1 and G2 *mTert*−/−; *Pot1b*−/− metaphase spreads revealed progressive telomere shortening and increased number of telomere free chromosome ends (Fig. 3f, g; Supplementary

**Fig. 3 Accelerated telomere shortening in _Pot1b_⁻/⁻; _Tert_⁻/⁻ cells. a** Arrow points to the targeting site of CRISPR/Cas9 guide RNA of mouse _mTert_ cDNA. TEN: Telomerase Essential N-terminal domain. TRBD: Telomerase RNA Binding Domain. RT: Reverse Transcriptase domain. CTE: C-Terminal Extension domain. **b** Detection of telomerase activity in G3 _Pot1b_⁻/⁻ parental cells and _Pot1b_⁻/⁻; _Tert_^hypo clone #44 by the TRAP assay. 2000 cells were used in each reaction. Numbers indicate TRAP activity, with 1.0 set for G3 _Pot1b_⁻/⁻ samples. INC: internal control. **c** PNA-FISH of G3 _Pot1b_⁻/⁻ sarcoma and _Pot1b_⁻/⁻; _Tert_^hypo clone #44 reconstituted with GFP or POT1b^WT. Scale bar: 5 μm. **d** Distribution of telomere length in (**c**) quantified by Q-FISH. At least 40 metaphases were analyzed per genotype in three independent experiments. Median telomere length are indicated. **e** Detection of G-overhang (native gel) and total telomere lengths (denature gel) in cells of the indicated genotypes by TRF Southern. *: DNA band used for quantification. EtBr: Ethidium Bromide staining shows equal loading of genomic DNA. Molecular weight markers are indicated. *: DNA band used for quantification. Numbers indicate relative G-overhang and total telomere signals, with telomere signals set to 1.0 for cells expressing GFP. **f** Telomere lengths in primary MEFs of the indicated genotypes measured by Q-FISH and percentage of telomere-free ends. A minimum of 40 metaphases were analyzed per genotype. At least 1500 chromosomes were scored for telomere signals and data show the mean ± s.d. from different cell lines with same genotyping. _Pot1b_⁻/⁻; _Tert_+/+ 3 cell lines; _Pot1b_⁻/⁻; _Tert_+/− 2 cell lines and _Pot1b_+/−; _Tert_⁻/⁻ 2 cell lines. **g** PNA-FISH images of SV40-immortalized _Pot1b_⁻/⁻ or _Pot1b_⁻/⁻, _Tert_⁻/⁻ MEFs expressing GFP or POT1b^WT. Scale bar: 5 μm. **h** Quantification of relative telomere lengths in (**g**). A minimum of 40 metaphases were analyzed per genotype. Box plots show the interquartile range (25% to 75%) and median from two independent experiments. p-values are shown and generated from unpaired Student's _t_ test.

Fig. 6a). _mTert_+/−; _Pot1b_⁻/⁻ and _mTert_⁻/⁻; _Pot1b_+/− MEFs also exhibited telomeres that are significantly shorter than those found in WT MEFs, suggesting that POT1b cooperates with telomerase to help maintain telomere length (Fig. 3f, g; Supplementary Fig. 6a). While reconstituted POT1b^WT readily localized to the telomeres of G1 and G2 _mTert_⁻/⁻; _Pot1b_⁻/⁻ MEFs (Supplementary Fig. 6b, c), telomere lengthening and a reduction of the number of telomere-free chromosome fusions were not observed in multiple independent G1 and G2 _mTert_⁻/⁻; _Pot1b_⁻/⁻ MEFs expressing POT1b^WT (Fig. 3g, h, Supplementary Fig. 6d, e). These results indicate that POT1b-mediated telomere elongation is completely dependent on the presence of telomerase.

**POT1b promotes telomerase recruitment to telomeres**. Biochemical evidence suggests that the human POT1-TPP1 complex stimulates telomerase activity in vitro[14]. We postulate that POT1b might facilitate TPP1's recruitment of telomerase to telomeres. To address this question, we used a telomerase recruitment assay that detects the co-localization of human telomerase to telomeres[33]. We co-expressed mTERT, _hTR_, mTPP1 and POT1a or POT1b in G3 _Pot1b_⁻/⁻ sarcomas and assayed for the presence of telomerase on telomeres by co-immunofluorescence and immunofluorescence-FISH (Supplementary Fig. 7a)[16]. Human telomerase RNA (_hTR_) interacts robustly with the protein component of mouse telomerase (mTERT) in vitro and localizes to mouse telomeres in the presence of mouse Flag-POT1a/b and HA-TPP1 (Supplementary Fig. 7b, Fig. 4a). In the presence of HA-TPP1, _hTERC_ RNA FISH analysis revealed that expressing either Flag-POT1b^WT or Flag-POT1b^ΔCST resulted in ~150% increase in the level of telomerase localized to telomeres over GFP control, as indicated by the co-localization of _hTERC_ with Flag-TPP1 (Fig. 4a, b). POT1b^WT also promoted the recruitment of telomerase to telomeres in SV40-immortalized _Pot1b_⁻/⁻ MEFs (Supplementary Fig. 7c). In contrast, reconstituted Flag-POT1b^F62A or Myc-POT1a^WT performed no better than the GFP control in recruiting telomerase to telomeres. These results reinforce our observations that POT1b's interaction with the 3′ G-overhang is essential for telomerase recruitment to telomeres and telomere elongation. In support of this notion, TRAP assays revealed that expressing POT1b^WT or POT1b^ΔCST, but not POT1b^F62A or POT1a^WT, in G3 _Pot1b_⁻/⁻ sarcomas significantly increased telomerase activity (Supplementary Fig. 7d, e).

Previous results suggest that human POT1 brings TPP1 close to the 3′ telomere end, where TPP1's TEL patch interacts directly with TERT to stimulate telomerase activity and processivity[13,16]. Since co-expression of TPP1 with POT1a and POT1b increases the stability of the POT1a/b proteins[34], we generated a series of expression constructs where POT1a^WT, POT1a^F62A, POT1b^WT

and POT1b^F62A are physically tethered to TPP1 via a 10 amino acid flexible linker (Supplementary Fig. 8a–c). POT1a^WT-TPP1, POT1b^WT-TPP1, POT1a^F62A-TPP1 and POT1b^F62A-TPP1 all efficiently localized to telomeres (Supplementary Fig. 8d). POT1a^WT-TPP1 behaved similarly as POT1a^WT in its ability to prevent the induction of a robust DNA damage response at telomere (Supplementary Fig. 8e), while POT1b^WT-TPP1 recruited STN1 (by inference the entire CST complex) to telomeres as efficiently as POT1b^WT (Supplementary Fig. 8f). However, when reconstituted in _Pot1b_⁻/⁻ sarcomas, only POT1b^WT-TPP1 was able to promote telomerase recruitment to telomeres and elicit telomere elongation (Fig. 4c–e; Supplementary Fig. 7c). We next asked whether POT1b is required to promote telomerase processivity. All four tethered constructs were co-expressed in 293T cells with mTERT and _hTerc_ and lysates harvested to perform a direct telomerase primer extension assay. Compared to cells expressing POT1a^F62A-TPP1 and POT1b^F62A-TPP1, expression of either POT1a^WT-TPP1 or POT1b^WT-TPP1 increased telomerase processivity, as indicated by the appearance of higher molecular weight products (Fig. 4f). Since telomerase processivity appears identical in the presence of either POT1a-TPP1 or POT1b-TPP1, our results suggest that both POT1a and POT1b are equally proficient to promote telomerase processivity in this in vitro assay.

**The POT1b OB-folds and C-terminus are required for telomerase recruitment to telomeres**. We postulate that POT1b cooperates allosterically with TPP1 to enhance telomerase recruitment to telomeres. To address this hypothesis, we generated a series of POT1a/POT1b chimeras to ask whether the POT1b N-terminal OB folds (amino acids 1-350) or its C-terminus (aa 351-640) is required to mediate telomerase recruitment to telomeres (Fig. 5a and Supplementary Fig. 9a). Expressing either the POT1a^1-350-POT1b^351-640 chimeric protein (abbreviated POT1ab) or the POT1b^1-350-POT1a^351-640 (abbreviated POT1ba) in G3 _Pot1b_⁻/⁻ sarcomas was unable to completely restore CST recruitment to telomeres (Supplementary Fig. 9b). However, only POT1ab was able to increase telomerase recruitment to telomeres by ~2-fold, similar to what we observed with POT1b^WT expression (Fig. 5b). Both telomere Q-FISH and TRF Southern analysis revealed that the POT1ab chimera induced telomere elongation by ~1.5 fold over GFP control (Fig. 5c, d). These results suggest that POT1b's C-terminal TPP1 interacting domain is essential for telomerase recruitment and telomere length elongation. They further reinforce our observations that the CST complex is not required for telomere elongation.

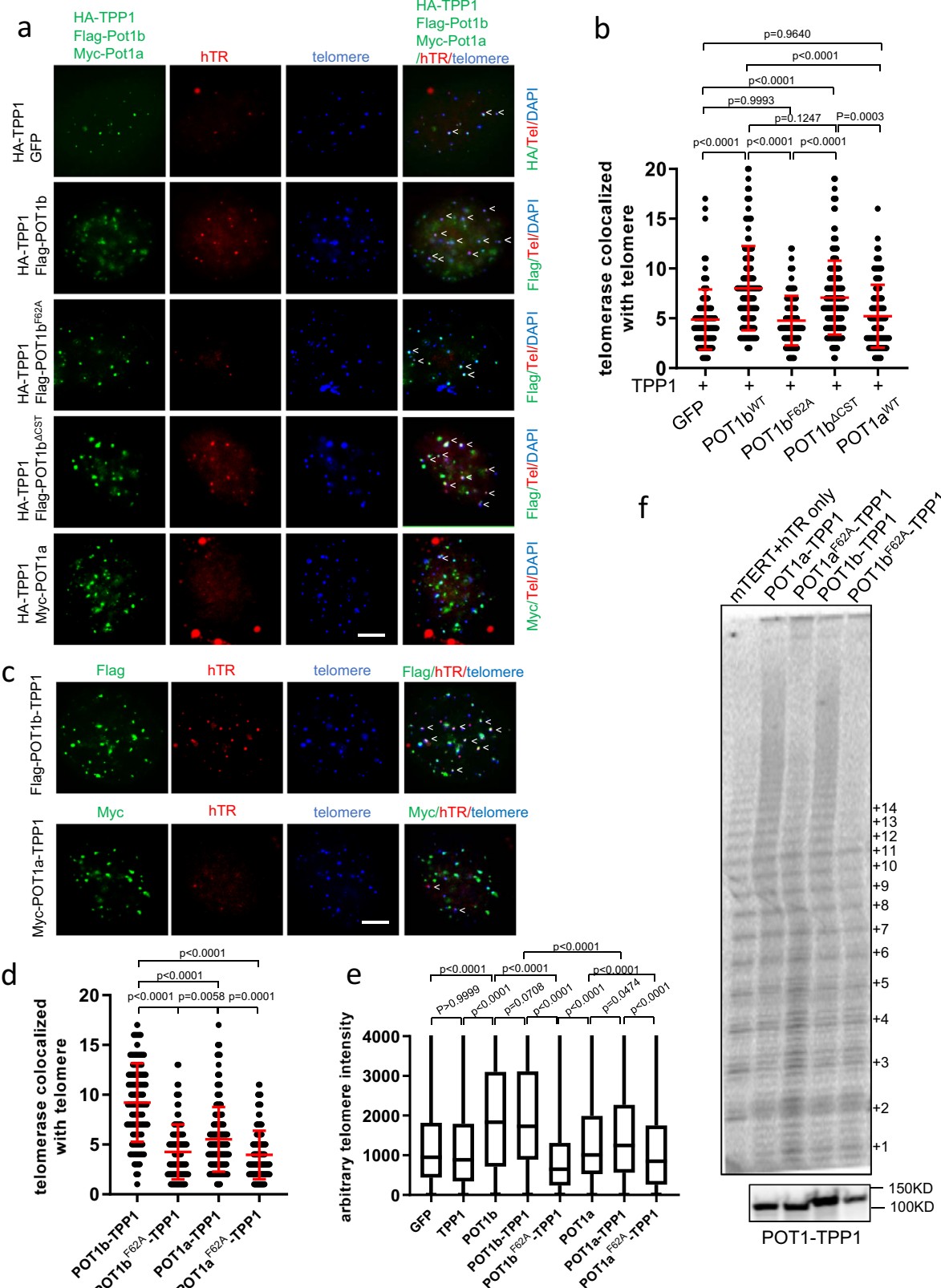

To understand why POT1a cannot promote telomerase recruitment to telomeres, we examined the cell-cycle-dependent localization of epitope-tagged POT1a, POT1a-TPP1, POT1b, and POT1b-TPP1 stably reconstituted in WT MEFs. Using the Fucci system to examine the cell-cycle profile of individual cells[35] (Supplementary Fig. 9c), we found that

~80% of cells in the S/G2 phase of the cell cycle displayed POT1b or POT1b-TPP1 localized to telomeres. In contrast, only 20% of cells in the G1 phase of the cell cycle displayed POT1b or POT1b-TPP1 localization to telomeres (Fig. 5e, f). The expression of POT1b-TPP1 significantly increased the number of STN1 foci on telomeres (Fig. 5g, h). Examination of POT1a

**Fig. 4 POT1b promotes telomerase recruitment to telomeres. a** Co-localization of hTR, HA-mTPP1, Flag-POT1b or Myc-POT1a on telomeres in G3 *Pot1b*$^{-/-}$ sarcomas (white arrows). HA-mTPP1, Flag-POT1b or Myc-POT1a (FITC, green) were detected by immunostaining with anti-HA, anti-Flag or anti-Myc antibodies. hTR RNA was detected by hybridization with Cy5-hTR cDNA probes (red) and telomeres visualized by hybridization with PNA probe Cy3-OO-(CCCTAA)$_4$ (blue). Scale bar: 5 μm. **b** Quantification of co-localization of hTR with telomeres in G3 *Pot1b*$^{-/-}$ sarcomas expressing the indicated DNAs in (**a**). Data show the mean ± s.d. from three independent experiments. At least 500 nuclei were analyzed per genotype. p-values are shown and generated from one-way ANOVA analysis followed by Tukey's multiple comparison. **c** Co-localization of hTR with POT1b-TPP1 or POT1a-TPP1 on telomeres in G3 *Pot1b*$^{-/-}$ sarcoms (white arrows). Scale bar: 5 μm. **d** Quantification of co-localization of hTR with telomeres in G3 *Pot1b*$^{-/-}$ sarcomas expressing the indicated DNAs in (**c**). Data show the mean ± s.d. from three independent experiments. At least 500 nuclei were analyzed per genotype. p-values are shown and generated from one-way ANOVA analysis followed by Tukey's multiple comparison. **e** Quantification of telomere lengths by Q-FISH in G3 *Pot1b*$^{-/-}$ sarcomas expressing the indicated DNAs. At least 30 metaphases were analyzed per genotype. Data are from three independent experiments and box plots show the interquartile range (25 to 75%) and median from three independent experiments. p-values are shown and generated from one-way ANOVA analysis followed by Tukey's multiple comparison. **f** Telomerase processivity assay. 293T cells stably expressing mTERT and hTR were reconstituted with POT1a-TPP1, POT1b-TPP1 or the indicated mutants. The lysates were then subjected to a direct telomerase activity assay to examine telomerase processivity in the presence of POT1a-TPP1 or POT1b-TPP1. Bottom panel: Western blot indicating equal expression of POT1-TPP1 proteins used in the processivity assays.

or POT1a-TPP1's telomere localization revealed no cell-cycle preference (Fig. 5e, f). In addition, the expression of POT1a-TPP1 significantly repressed the localization of both endogenous STN1 and telomerase to telomeres (Fig. 5h, i). Since STN1 is recruited to telomeres as part of the CST complex by POT1b, our data suggest that POT1a-TPP1 expression also represses POT1b and/or CST localization to telomeres.

**Mutational analyses of Pot1b-TPP1 interaction.** Using X-ray crystallography, we previously solved the C-terminus structure of human POT1 complexed with the hPOT1 binding motif of hTPP1[36]. The C-terminus of hPOT1 (amino acid residues 320-634) contains a third OB-fold bearing a holiday junction like resolvase (HJRL). Both OB3 and HJRL form grooves that interact with two α-helices and a 3$_{10}$ helix of hTPP1's hPOT1 binding motif (PBM: hTPP1$_{266-320}$)[36,37]. Importantly, several human cancer mutations map in this region, suggesting that disrupting hPOT1-hTPP1 interaction promotes tumorigenesis. Since telomerase is recruited to telomeres by hTPP1, we hypothesized that residues in the OB3 and HJRL domains of POT1b interact with mTPP1 to promote telomerase recruitment, and that these corresponding amino acids are missing in POT1a. Because hPOT1 possesses functions of both POT1a and POT1b, we reasoned that like POT1b, it should also contain amino acids that interact with mTPP1 to recruit telomerase to telomeres. To test this hypothesis, we examined the crystal structure of the hPOT1 C-terminal-hTPP1 complex, as well as the in silico structures of POT1a and POT1b complexed with mTPP1. Using the hPOT1-hTPP1 structure as a guide, we uncovered a cluster of POT1b amino acids, namely, D421, D426, and E428 predicted to form both ionic interactions and hydrogen bonds with mTPP1's R180 (Fig. 6a, b). In hPOT1, the corresponding residue N415, D420 and K422 are predicted to form hydrogen bonds with hTPP1 Q268. In contrast, the corresponding amino acids A421, Y426, and K428 in POT1a are not predicted to form any interaction with mTPP1 R180. Other amino acids predicted to interact with mTPP1 residues include POT1b P381, which is predicted to form hydrophobic interactions with mTPP1 W205, and POT1a V551, predicted to form hydrophobic interaction with mTPP1 F218 (Fig. 6a, b). Corresponding residues S381 in POT1a and T551 in POT1b are predicted to have reduced interaction with mTPP1 (Fig. 6a, b).

We used site-directed mutagenesis to convert the amino acids in POT1b predicted to interact with mTPP1 into amino acids encoded by POT1a, either singly or in groups, to generate 5 POT1b constructs bearing POT1a amino acids (Fig. 6c). For example, POT1b residues D421, D426 and E428 were mutated into A421, Y426 and K428. All constructs co-expressed robustly with mTPP1 in 293T cells (Fig. 6d). Co-IP experiments revealed

that other than POT1b$^{LS}$, POT1b mutants bearing POT1a residues were unable to interact with mTPP1 as well as POT1b$^{WT}$ (Fig. 6d). Telomerase recruitment assays, Q-FISH and TRF Southern analyses revealed that only POT1b$^{WT}$ and the POT1b$^{LS}$ mutant were able to recruit telomerase to telomeres to promote telomere elongation (Fig. 6e, f; Supplementary Fig. 10a, b, d). In particular, the POT1b$^{AYK}$ mutant was indistinguishable from POT1a$^{WT}$ in its inability to recruit telomerase to telomeres to promote telomere elongation (Fig. 6e, f). Compared to POT1b$^{WT}$, all other POT1b mutants examined displayed significantly reduced ability to recruit telomerase to telomeres and were unable to elongate telomeres as well as POT1b$^{WT}$, although all were better than POT1a$^{WT}$ in those regards. These results suggest that POT1b residues D421, D426, E428 in the HJRL domain are essential for interaction with TPP1 to promote telomerase recruitment and concomitant telomere elongation.

We next introduced POT1b amino acids D421, D426, E428, K376, P381 and T551 into POT1a, either singly or in groups, to generate 5 POT1a constructs bearing POT1b amino acids (Fig. 6c). All five POT1a mutants interacted with TPP1 as well as POT1a$^{WT}$ (Fig. 6g). Remarkably, POT1a$^{DDE}$ was able to recruit telomerase to telomeres as well as POT1b$^{WT}$ (Fig. 6h). POT1a$^{DDE}$ expression also increased telomere length, although not as well as POT1b$^{WT}$ (Fig. 6i; Supplementary Fig. 10c). These results confirm that the POT1b C-terminus HJRL domain amino acids D421, D426 and E428, predicted to form ionic interactions and hydrogen bonds with mTPP1's R180, are required for POT1b-mediated telomerase recruitment to telomeres and telomere elongation.

To rule out the possibility that endogenous POT1a in G3 *Pot1b*$^{-/-}$ cells might reduce the impact of the reconstituted POT1a mutants in promoting telomere elongation, we reconstituted GFP, POT1a$^{WT}$ or POT1a$^{DDE}$ into *MMTV-Cre; POT1a*$^{\Delta/\Delta}$; *p53*$^{\Delta/\Delta}$ mammary tumor cell lines[34]. Expression of POT1a$^{WT}$ and POT1a$^{DDE}$ almost completely repressed the increased TIFs observed in these cells (Supplementary Fig. 11a–d). In contrast to POT1a$^{WT}$, whose expression inhibited telomerase recruitment to telomeres and promoted telomere shortening, expression of POT1a$^{DDE}$ promoted both telomerase recruitment to telomeres and telomere elongation of the shortest telomeres (Supplementary Fig. 11e, f; Fig. 6j, k). These results further confirm that amino acids D421, D426 and E428 in POT1b are essential to stimulate telomerase recruitment to telomeres to promote telomere elongation.

**POT1b enhances TPP1 TEL patch-dependent telomeres recruitment.** The TPP1 TEL patch is required for both telomerase recruitment to telomeres and for high-processivity telomere synthesis[16]. Using tethered POT1b-TPP1 constructs, we asked whether POT1b's ability to recruit telomerase to telomeres

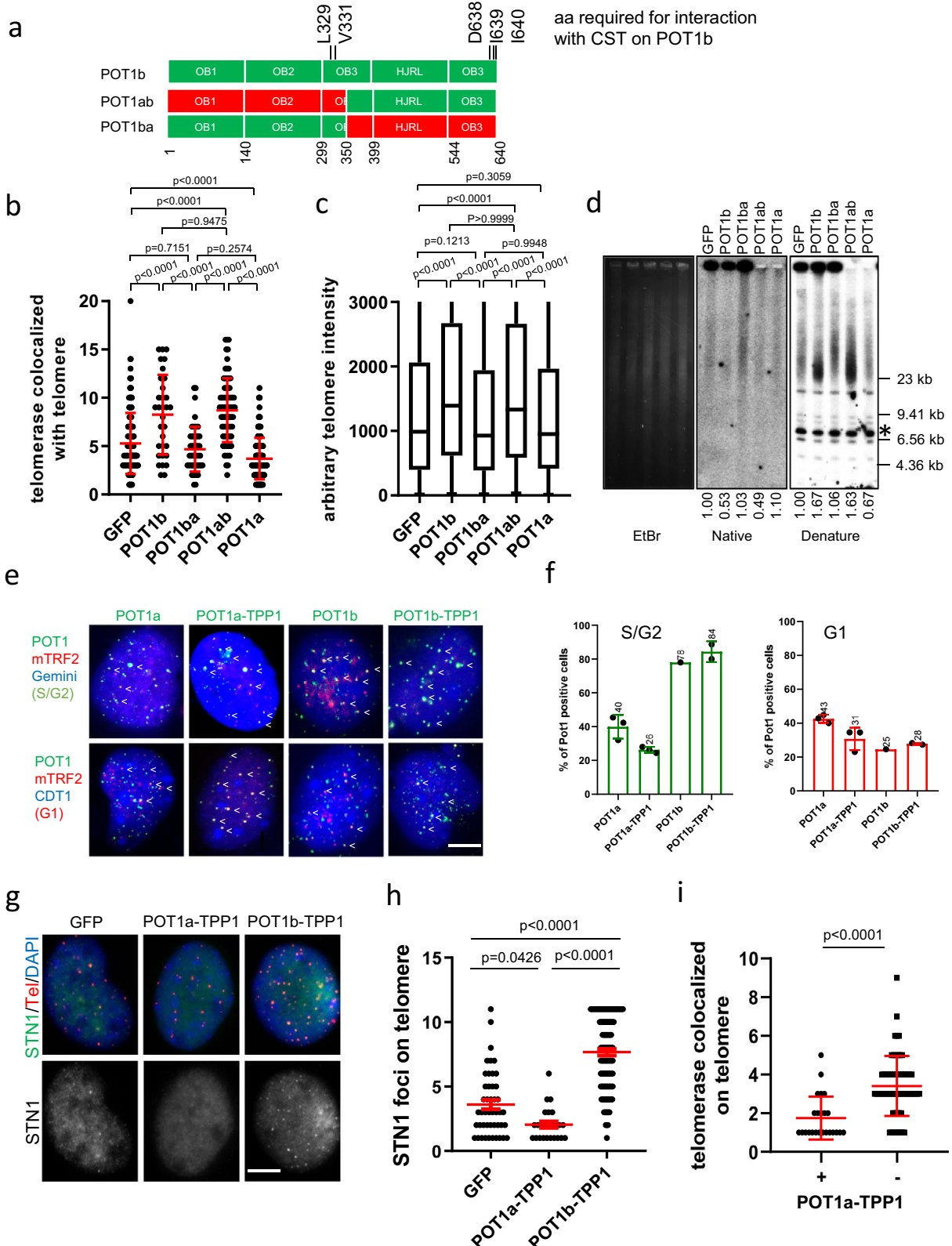

is dependent on TPP1's TEL patch. Expressing POT1b$^{WT}$-TPP1$^{WT}$ in G3 *Pot1b$^{-/-}$* sarcoma cell lines resulted in significantly increased telomerase recruitment to telomeres and concomitant telomere lengthening over GFP control (Fig. 7a, b). In contrast, expressing POT1b$^{WT}$-TPP1$^{ΔRD}$ (TPP1 lacking amino acids 159–246 that interacts with POT1 proteins), POT1b$^{AYK}$-TPP1$^{WT}$ or POT1b$^{LSV}$-TPP1$^{WT}$ did not result in telomerase

recruitment nor telomere lengthening. These results reinforce our observations suggesting that physical contact between POT1b's DDE residues and TPP1 is required for POT1b-mediated telomerase recruitment. Expressing POT1b$^{WT}$-mTPP1$^{ΔK82}$ (mTPP1 with a TEL patch mutation unable to interact with telomerase) also failed to recruit telomerase or lengthen telomeres (Fig. 7c, d; Supplementary Fig. 12a–c). While both POT1b$^{WT}$-TPP1$^{ΔK82}$ and

**Fig. 5 The POT1b C-terminal is required for the elongation of telomere. a** Schematic of POT1b, POT1ab and POT1ba constructs. All 5 amino acids in POT1b required to interact with CTC1 are shown. **b** Quantification of co-localization of hTR with telomeres in G3 *Pot1b*$^{-/-}$ sarcomas expressing the indicated DNAs. Data show the mean ± s.d. from three independent experiments. p-values are shown and generated from one-way ANOVA analysis followed by Tukey's multiple comparison. **c** Quantification of telomere lengths in G3 *Pot1b*$^{-/-}$ sarcomas expressing the indicated DNAs. At least 40 metaphases were analyzed per genotype. Box show the interquartile range (25 to 75%) and median from three independent experiments. p-values are shown and generated from one-way ANOVA analysis followed by Tukey's multiple comparison. **d** TRF Southern blot to detect G-overhangs (native) and total telomeres (denature) in G3 *Pot1b*$^{-/-}$ sarcomas expressing the indicated DNAs. *: DNA band used for quantification. EtBr: Ethidium Bromide staining reveals equal loading of genomic DNA. Numbers indicate relative G-overhang and total telomere signals, with telomere signals set to 1.0 for cells expressing GFP. Molecular weights are indicated. **e** WT MEFs expressing cell-cycle sensors Geminin (blue) or CDT1 (blue) were reconstituted with the indicated POT1 proteins (green). Endogenous TRF2 marked telomeres (red). White arrows indicate POT1/TRF2 co-localization. Scale bar: 5 μm. **f** Localization of POT1a, POT1a-TPP1, POT1b and POT1b-TPP1 on telomeres during S/G2 (Geminin positive, left panel) and during G1 (CDT1 positive, right panel). Data show the mean ± s.d. from three independent experiments. At least 200 nuclei were analyzed for each genotype. **g** Endogenous STN1 localized to telomere in wild-type MEFs expressing POT1a-TPP1 or POT1b-TPP1. STN1 was immunostained with anti-STN1 antibody (green) and telomeres were detected by PNA FISH with Cy3-OO-(CCCTAA)$_4$ (red). Nuclei were stained with DAPI (blue). Scale bar: 5 μm. **h** Quantification of (**g**). Data show the mean ± s.d. from three independent experiments. 200 nuclei were analyzed in two independent experiments. p-values are shown and generated from one-way ANOVA analysis followed by Tukey's multiple comparison. **i** POT1a-TPP1 expression represses telomerase recruitment to telomeres. Quantification of telomerase localization to telomere in WT MEFs expressing (+) POT1a-TPP1 or vector control (−). At least 200 nuclei were analyzed for each group and data show the mean ± s.d. from two independent experiments. p-values are shown and calculated from unpaired Student's *t* test.

POT1b$^{F62A}$-TPP1$^{\Delta K82}$ are equally impotent in terms of telomerase recruitment to telomeres, Q-FISH revealed that cells expressing POT1b$^{F62A}$-TPP1$^{\Delta K82}$ display significantly shorter telomere lengths (Fig. 7d). This result suggests that in cells expressing POT1b$^{F62A}$-TPP1$^{\Delta K82}$, endogenous POT1a can now interact with the 3′ overhang, preventing telomerase access to telomeres. Taken together, our data reveal that POT1b's ability to promote telomerase recruitment requires specific contacts with TPP1 to enhance TPP1's ability to recruit telomerase to telomeres through the TPP1 TEL patch.

## Discussion

Telomere end replication requires telomerase to base pair with the 3′ ss overhang of telomere DNA. However, this ss G-overhang must also be protected from engaging in the activation of an ATR-dependent DNA damage signal that would otherwise elicit illegitimate homology directed repair, generating aberrant chromosome fusions that might be cancer promoting[12,21]. Results presented here suggest that POT1 proteins play crucial roles in both telomere end protection and end replication. Although hPOT1 clearly possesses telomere end protective functions[38], conflicting biochemical and genetic evidence reveal that it functions both as a negative and a positive regulator of telomerase activity at telomeres[9,18]. Fortuitously, the two mouse POT1 proteins possess separation of functions found in hPOT1, allowing us to dissect the roles that hPOT1 plays in telomere length maintenance. In this report, we show that POT1b interacts with TPP1 to promote telomerase recruitment to telomeres to mediate telomere G-strand elongation as well as C-strand fill-in through its recruitment of the CST complex to telomeres. In contrast, POT1a functions to repress activation of a DDR at telomeres and negatively regulates telomerase access to telomeres. Our work provides mechanistic insight into how POT1a and POT1b regulate telomere end protection and telomere end replication, respectively.

The very short telomeres observed in *G3 Pot1b*$^{-/-}$ sarcomas make it a highly sensitive system to examine the impact of POT1a and POT1b on telomere maintenance. Expression of POT1b$^{WT}$, but not POT1a$^{WT}$, resulted in increased recruitment of telomerase to telomeres and concomitant telomere elongation. The POT1b$^{F62A}$ mutant was unable to elongate telomeres, suggesting that POT1b's N-terminal OB-folds are required to interact with the ss G-rich telomeric overhang to facilitate telomerase recruitment. In support of this notion, domain swap experiments

revealed that the N-terminal OB-folds of POT1a functioned as well as those from POT1b in interacting with the 3′ G-overhang to promote telomerase recruitment to telomeres (Fig. 5). These observations are in agreement with previous reports showing that hPOT1 functions to tether hTPP1 close to the 3′ end of telomeric DNA, enabling the hTPP1 TEL patch and NOB region to interact with telomerase to recruit it to telomere ends[13,16,18,39,40].

The purified hPOT1-hTPP1 heterodimer has a 3-fold higher affinity for telomeric ssDNA than hPOT1 alone and promotes increased telomerase processivity in vitro[13,18]. These results suggest that hPOT1 plays important roles in enhancing telomerase function at telomeres. The hPOT1 C-terminus contains a third OB fold with a HJRL domain inserted within it, and both domains interact with hTPP1's PBM[36,37]. Residues N415, D420 and K422 in the hPOT1 HJRL domain form hydrogen bonds with Q268 in hTPP1[36]. The corresponding amino acid residues in POT1b, D421, D426 and E428, are predicted to form both ionic interactions and hydrogen bonds with mTPP1$^{R180}$. Changing these amino acids to corresponding ones found in POT1a abolished POT1b's ability to promote telomerase recruitment. Intriguingly, incorporating these three POT1b amino acids into POT1a rendered POT1a$^{DDE}$ as effective as POT1b$^{WT}$ in recruiting telomerase to telomeres and enhancing telomere elongation (Fig. 6), suggesting that specific contacts between hPOT1 and POT1b's HJRL with TPP1$^{PBM}$ are essential to promote telomerase recruitment to telomeres. To test the hypothesis that POT1b's interaction with mTPP1 imparts allosteric activation of the mTPP1 TEL patch to enhance telomerase recruitment, we tethered POT1b$^{WT}$ or POT1b mutants to mTPP1$^{WT}$ or the mTPP1$^{\Delta K82}$ mutant. Only POT1b$^{WT}$-mTPP1$^{WT}$ supported telomerase recruitment and telomere elongation, suggesting that direct POT1b interaction with mTPP1 and an intact mTPP1 TEL patch are both essential for POT1b-mediated allosteric activation of mTPP1 to increase the recruitment of telomerase to telomeres (Fig. 7).

Structural analysis of the hPOT1-hTPP1-ss telomeric DNA complex using small-angle X-ray scattering revealed that the hPOT1-hTPP1-ss DNA complex adopts an elongated V-shaped topology. The two N-terminal hPOT1$^{OB1}$, hPOT1$^{OB2}$ -folds are located at one end of the complex in contact with the ss DNA, while the hPOT1$^{OB3}$, hTPP1$^{PBM}$ and the hTPP1$^{OB-TEL patch}$ are located at the other end[36]. We postulate that POT1b$^{OB3}$ interaction with mTPP1$^{PBM}$ alters the mTPP1$^{OB-TEL patch}$ conformation, enhancing its direct interaction with mTERT to recruit telomerase to

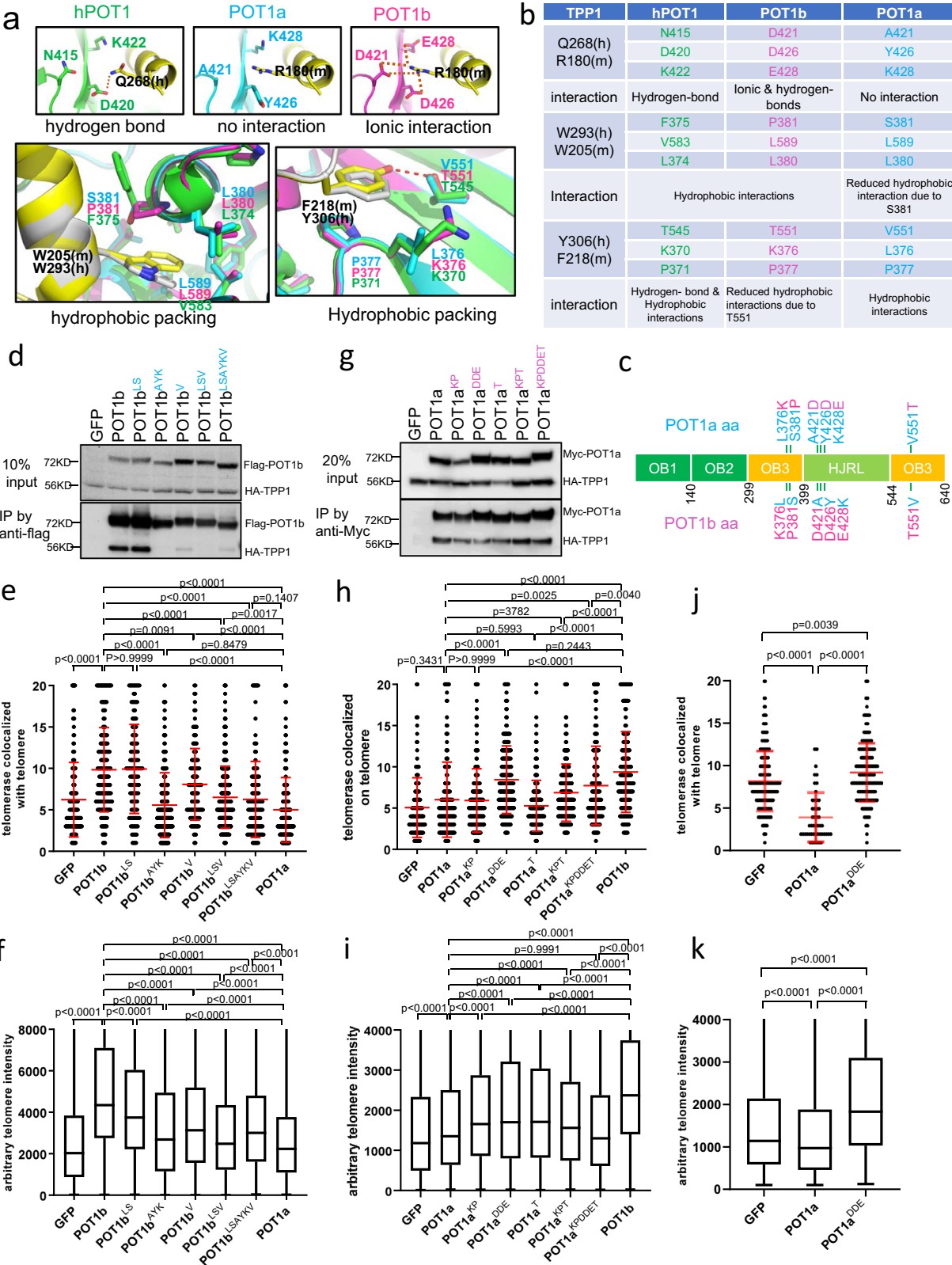

telomeres. We also cannot rule out the possibility that POT1b$^{DDE}$ forms a direct interaction with mTERT. In this scenario, the POT1b$^{OB3}$ would provide another surface that cooperates with mTPP1$^{OB-TEL\ patch}$ to interact with mTERT to enable telomerase recruitment and telomere extension.

To allow telomerase access the 3′ G-overhang, the N-terminal mPOT1b$^{OB1,\ OB2}$ domains must bind to internal TTAGGG

repeats, enabling POT1b-mTPP1 to position telomerase to the very end of the 3′ G-overhang to promote telomere elongation. hPOT1-hTPP1's linear conformation also enables it to control access to the 3′ G-overhang. While both POT1a and POT1b bind to the ss 3′G-overhang with equal affinity to prevent RPA binding and activation of an ATR-dependent DDR[41], POT1a is better at suppressing a DDR at telomeres than POT1b. We show that

**Fig. 6 Identification of unique amino acids required for POT1b dependent telomere elongation. a** Interaction interfaces between POT1 and TPP1. POT1: green; POT1a: cyan; POT1b: magenta; hTPP1/mTPP1: black. Top three panels show the POT1/POT1a/POT1b interactions around residues Q286hTPP1/R180mTPP1. The bottom left panel shows hydrophobic interactions around residues W293hTPP1/W205mTPP1, and the bottom right panel shows hydrophobic interactions around residues Y306hTPP1/F218mTPP1. **b** Table of amino acids from (**a**) predicted to form direct interactions between POT1 and TPP1. **c** Schematic of point mutations generated to change specific residues in POT1a or POT1b. **d** Co-IP assay reveal that most POT1b mutants interacts poorly with TPP1. **e** Quantification of co-localization of hTR with telomeres in G3 $Pot1b^{-/-}$ sarcomas reconstituted with the indicated POT1a and POT1b DNAs. 500 nuclei were analyzed per genotype. Data show the mean ± s.d. from three independent experiments. **f** Quantification of telomere lengths by Q-FISH in G3 $Pot1b^{-/-}$ sarcomas expressing POT1a or POT1b constructs. 30 metaphases were analyzed per construct. Box show the median and interquartile range (25% to 75%) from three independent experiments. **g** Co-IP assay revealed that POT1a mutants interact robustly with TPP1. **h** Quantification of co-localization of hTR with telomeres in G3 $Pot1b^{-/-}$ sarcomas reconstituted with the indicated POT1a and POT1b constructs. 500 nuclei were analyzed per genotype. Data show the mean ± s.d. from three independent experiments. **i** Quantification of telomere lengths by Q-FISH in G3 $Pot1b^{-/-}$ sarcomas expressing POT1a or POT1b DNAs. 30 metaphases were analyzed per construct. Box show the median and interquartile range (25% to 75%) from three independent experiments. **j** Quantification of co-localization of hTR with telomeres in $MMTV\text{-}Cre; p53^{\Delta/\Delta}; Pot1a^{\Delta/\Delta}$ cells reconstituted with POT1a constructs. 500 nuclei were analyzed per genotype. Data show the mean ± s.d. from three independent experiments. **k** Q-FISH quantification of telomere length in $MMTV\text{-}Cre; p53^{\Delta/\Delta}; Pot1a^{\Delta/\Delta}$ tumor cells reconstituted with the indicated DNAs. 40 metaphases were analyzed. Box show the median and interquartile range (25–75%) from three independent experiments. For panels (**e**, **f**, **h–k**), p-values are shown and generated from one-way ANOVA analysis followed by Tukey's multiple comparison.

POT1a's ability to outcompete POT1b for access to the 3′ G-overhang, preventing CST and telomerase accumulation on telomeres might be cell cycle regulated (Fig. 5). POT1a localizes to telomeres throughout the cell cycle. However, during S/G2, accumulation of POT1b on telomeres increase by ~2-fold, likely allowing POT1b to out compete POT1a at the 3′ G-overhang to promote telomerase recruitment. The ability of POT1b to regulate telomerase extension of the G-strand and its recruitment of the CST complex to telomeres to modulate C-strand fill-in is reminiscent of the *Arabidopsis* POT1a protein, which stimulates telomere elongation by telomerase and binds STN1 and CTC1[42].

Recent genome-wide cancer sequencing efforts revealed germline and somatic hPOT1 mutations in diverse cancer types, including chronic lymphocytic leukemia[43], melanoma[44,45], glioma[46] and angiosarcoma[47]. These results reveal that hPOT1 is the most highly mutated shelterin component in human cancers. Cancer-associated hPOT1 mutations cluster mainly within the first two OB-folds, disrupting hPOT1's ability to bind the telomeric overhang and leads to telomere elongation[34]. Several hPOT1 cancer mutations in the C-terminal OB-3 have been reported to disrupt hPOT1's interaction with hTPP1, resulting in increased DNA damage response and telomere length extension[34,36,37]. These results suggest that the majority of hPOT1 cancer mutations diminish hPOT1's ability to repress telomerase access to the 3′ overhang. Based on our results, we speculate that some of the telomere elongation phenotypes observed in cancer cells bearing hPOT1 OB-3 mutations might be due to enhanced telomerase recruitment to telomeres, leading to increased telomere elongation, which would endow precursor cancer cells with a proliferative advantage. While it is still unclear how hPOT1 switches from its end protective functions to increased telomerase recruitment, the identification of unique amino acids between POT1a and POT1b will help us understand mechanistically how hPOT1 switches between these two states.

## Methods

**Generation of p53 $^{\Delta/\Delta}$; Pot1a$^{F/F}$; Pot1b$^{+/-}$ and p53 $^{\Delta/\Delta}$; Pot1a$^{F/\Delta}$; Pot1b$^{-/-}$ sarcoma cell lines and Pot1b$^{-/-}$; mTert$^{-/-}$ mouse model and MEFs.** *CAG-Cre$^{ER}$; p53$^{F/F}$; Pot1a$^{F/F}$; Pot1b$^{+/-}$* mice were generated from multiple-step cross-mating between *Pot1a$^{F/F}$* mice[12], *Pot1b$^{-/-}$* [17], *p53$^{F/F}$* mice and *CAG-Cre$^{ER}$* mice (from Jackson Laboratory of United States). Primary Mouse Embryonic Fibroblasts (MEFs) *CAG-Cre$^{ER}$; p53$^{F/F}$; mPot1a$^{F/F}$; mPot1b$^{+/-}$* and *CAG-Cre; p53$^{F/F}$; mPot1a$^{F/F}$; mPot1b$^{-/-}$* were isolated from embryos generated from cross-mating between *CAG-Cre$^{ER}$; p53$^{F/F}$; Pot1a$^{F/F}$; Pot1b$^{+/-}$* and *p53$^{F/F}$; Pot1a$^{F/F}$; Pot1b$^{+/-}$* mice. After treated with 4-hydroxy-Tamoxyfen (4-HT) or Adeno-Cre and kept passaging for 3 months, *p53* was completely deleted and Pot1a was partially removed in immortalized MEF (G0). G1 sarcoma cell lines were harvested from ICR-SCID mice sarcoma by subcutaneous injection of G0 MEF into 8-week ICR-SCID mice. G2 and G3 sarcoma cell lines were isolated from the repeated injection of G1 and G2 sarcoma cells into ICR-SCID mice. The injection procedure was described in detail in Supplementary figure 1A. ICR-SCID mice were generated by standard mating. *Pot1b$^{+/-}$; Tert$^{+/-}$* mice was generated from mating of *Pot1b$^{-/-}$* mice with *mTert$^{+/-}$* mice. *Pot1b$^{-/-}$; mTert$^{-/-}$* MEF was isolated from embryos by matings between *Pot1b$^{+/-}$; mTert$^{+/-}$* mice. All mice were maintained according to the IACUC-approved protocols of Yale University. All cells including primary MEF, G0 MEF and G1, G2 and G3 sarcomas were cultured in DMEM/high glucose media supplemented with 10% FBS. Mouse embryonic fibroblast cell lines *CAG-Cre$^{ER}$; mPot1b$^{-/-}$; mCTC1$^{F/F}$* were generated from the corresponding mice[26], immortalized with SV40-large T antigen (LT) and maintained in DMEM/high glucose media supplemented with 10% FBS. Human cell lines 293T was cultured in the same media.

**Plasmids, retrovirus and antibodies.** POT1a and POT1b mutants were generated by PCR and constructed in retrovirus expression vector pQCXIP-puro. The fusion protein POT1-TPP1 was linked by a 10 amino acids polyglycine spacer and a Flag or Myc tag was inserted at the N-terminus of POT1. Mouse telomerase expression vector HA-mTerT/ pMGIB (gift from Steven Artandi's lab) and human telomerase RNA expression vector pBABEpuro-U3-hTR-500 was purchased from addgene (#27666)[33]. Fucci-mKO-CDT-1 and Fucci-mAG-Geminin plasmids were gift from Miyawaki's lab[35]. All lentivirus or retrovirus was generated in 293T cells and infected target cells twice. Antibodies that recognize phosphorylated γH2AX (Millipore #05-636) was used for the DNA damage assays. Anti-epitope Tag antibodies were purchased from Sigma (anti-Flag #F3165 and anti-HA #A300-305A) or Millipore (anti-Myc #05-724). Anti-Tag antibody cross-linked agarose beads were purchased from Sigma (anti-Flag beads #A2220, Anti-Myc beads (#A7470) and anti-HA beads(#A2095). Anti-mTRF2 antibody was purchased from Millipore (#05-521).

**Disruption of mTert gene through CRISPR/Cas9.** Mouse telomerase guide RNA mTert sgRNA [5′-GCTACGGGAGCTGTCAC(PAM)-3′] was designed online (https://chopchop.rc.fas.harvard.edu/) and cloned into lentivirus CAS9 vector LentiCRISPRv2puro (Addgene #89290) and delivered into G3 $p53^{-/-}$; $Pot1a^{F/-}$; $Pot1b^{-/-}$ sarcoma cell by infection of lentivirus produced in 293T cells. Infected cells were selected by puromycin for several days and single clones were picked up and amplified. The targeted clones were screened by limited dilution and disruption of gene was confirmed by sequencing of Topo TA-cloned PCR products of targeting locus [PCR primers:5′- CTGCATGCTCCTGTCATAACTC-3′ and 5′-GACTCAACCATCAGTACAGGGG-3′].

**Telomere signal analysis by Telomere PNA-FISH and Image J.** Cells were treated with 0.5 μg/ml of Colcemid (Invitrogen) for 4 h before harvest. Chromosomes were fixed with 3% formamide and hybridized with a telomere PNA-FISH probe 5′-Cy3-OO-(CCCTAA)$_4$-3′ (PNAgene) as described[12]. DNA was counterstained with DAPI. Digital images were captured using NIS-Elements BR (Nikon) with a Nikon Eclipse 80i microscope utilizing an Andor CCD camera. The relative telomere signals were analyzed with ImageJ software (downloaded from Fiji).

**Immunofluorescence and fluorescent in situ hybridization.** Cells grown in 8 well chambers were fixed for 10 min in 2% (w/v) sucrose and 2% (v/v) paraformaldehyde at room temperature followed by PBS washes and permeabilized with 0.5% NP40 for 10 min. Cells were blocked for 1 h in blocking solution (0.2% (w/v) fish gelatin and 0.5% (w/v) BSA in 1× PBS). The cells were incubated with primary

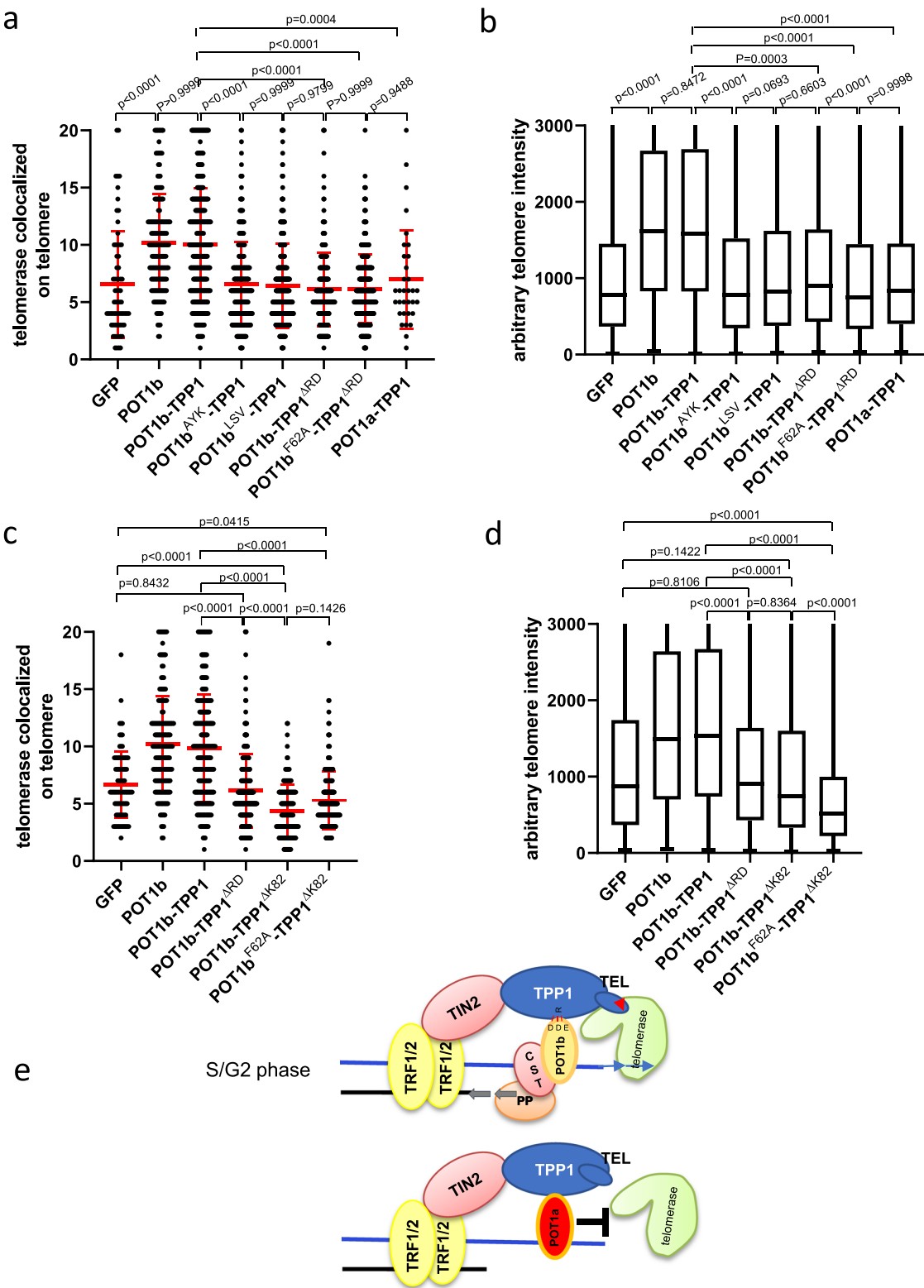

antibodies overnight at 4 °C. After 0.1% Triton-PBS washes, slides were incubated with the appropriate Alexa fluor secondary antibody for 1 h followed by washes in 1XPBS with 0.1% Triton. PNA-FISH was carried out as above using a PNA telomere probe 5′-Cy3-OO-(CCCTAA)$_4$-3′ (PNAgene). Digital were captured as above.

**Cell-cycle-dependent localization of POT1 in MEF.** Wild-type MEF was infected with retrovirus of POT1a or POT1b followed by infection of Fucci-CDT1 or Fucci-geminin lentivirus and maintained for 3 days. Then the cells were processed for immunostaining with anti-tag (Myc for POT1a and Flag for POT1b) and mTRF2 antibody. The cells were mounted under coverslips without DAPI mounting reagent before taking images. The distribution of POT1a or POT1b in wild-type MEF during different cell cycle were counted based on the fluorescent Tagged POT1 signals with or without fragments of CDT-1 or Geminin as G1 or S/G2 reporters, respectively[35].

**Detection of telomerase RNA (TR) by RNA-FISH in mouse cells.** G3 *Pot1b*$^{−/−}$ sarcoma cells or G0 *Pot1b*$^{−/−}$ MEF or in *MMTV-Cre; p53*$^{Δ/Δ}$; *Pot1a*$^{Δ/Δ}$ was co-

**Fig. 7 Both POT1b-TPP1 interaction and the TPP1 TEL patch are required to recruit telomerase to telomeres. a** Quantification of co-localization of hTR with telomeres in G3 $Pot1b^{-/-}$ sarcomas reconstituted with the indicated POT1b or tethered DNAs. 500 nuclei were analyzed per genotype. Data show the mean ± s.d. from three independent experiments. p-values are shown and generated from one-way ANOVA analysis followed by Tukey's multiple comparison. **b** Quantification of telomere lengths by Q-FISH in G3 $Pot1b^{-/-}$ sarcomas expressing POT1b or tethered DNAs. A minimum of 35 metaphases were analyzed per genotype. Box show the median and interquartile range (25% to 75%) from three independent experiments. p-values are shown and generated from one-way ANOVA analysis followed by Tukey's multiple comparison. **c** Quantification of co-localization of hTR with telomeres in G3 $Pot1b^{-/-}$ sarcomas reconstituted with the indicated POT1b or tethered DNAs. 500 nuclei were analyzed per genotype. Data show the mean ± s.d. from three independent experiments. p-values are shown and generated from one-way ANOVA analysis followed by Tukey's multiple comparison. **d** Quantification of telomere lengths by Q-FISH in G3 $Pot1b^{-/-}$ sarcomas expressing POT1b or tethered DNAs. At least 35 metaphases were analyzed per genotype. Boxes show the median and interquartile range (25% to 75%) from three independent experiments. p-values are shown and generated from one-way ANOVA analysis followed by Tukey's multiple comparison. **e** Schematic of how POT1b promotes telomerase recruitment to telomeres. See text for details.

infected with retrovirus expression mouse telomerase from vector HA-mTert/pMGIB and human telomerase RNA from vector pBABEpuro-U3-hTR-500. After double selection with 10 ug/ml of Blasticidin and 2 ug/ml of puromycin for several days, the super-mTert/hTR cells were established and used for telomerase recruitment assay. The super-mTert/hTR cells were co-infected with retrovirus expressing mTPP1 and wild-type or mutant POT1a or POT1b and maintained for 72 hrs. Following immunostaining with anti-Tag antibody, cells were incubated in prehybridization solution (0.1% Dextran sulfate, 1 mg/ml of BSA, 2xSSC, 50% formamide, 0.5 mg/ml spermidine DNA, 0.1 mg/ml *E.coli* tRNA, 1 mM RNase inhibitor VRC) at 37 °C for 1 h, then hybridized with combined PNA-FISH probe Cy3-OO-(CCCTAA)$_4$ and Cy5-hTR cDNA probes in prehybridization solution overnight at 37 °C. The washing conditions are the same as described for PNA-FISH. Digital were captured as above.

**Co-IP experiments**. To examine whether mutant POT1a or POT1b was able to interact with TPP1 in vitro, 293T cells were co-transfected with WT or mutant Myc-POT1a or Flag-POT1b and HA-mTPP1, cell lysates were incubated in TEB$_{150}$ buffer (50 mM Hepes pH7.3, 150 mM NaCl, 2 mM MgCl$_2$, 5 mM EGTA, 0.5% Triton-X-100, 10% Glycerol, proteinase inhibitors) for 6 h at 4 °C with anti-Tag antibody cross-linked agarose beads. After washing with TEB$_{150}$ buffer three times, the elution from beads was analyzed by immunoblotting.

**In vitro DNA-binding assay**. Streptavidin-sepharose beads (Invitrogen) coated with Biotin-Tel-G (TTAGGG)$_6$ were used for the ss DNA-binding assays. The coated beads were incubated with crude 293T cell lysates in TEB$_{150}$ buffer (50 mM Hepes, pH 7.3, 150 mM NaCl, 2 mM MgCl$_2$, 5 mM EGTA, 0.5% Triton-X-100, 10% glycerol and proteinase inhibitors) overnight at 4 °C. After washing with the same buffer, the beads were analyzed by immunoblot assay.

**TRF southern blotting**. A total of $1 \times 10^6$ cells were suspended in PBS, 1:1 mixed with 1.8% agarose in 1×PBS and cast into plugs. The plugs were digested at 55 °C for 2 days with 1 mg/ml proteinase K (Roche) in 10 mM sodium phosphate (pH7.2) and 0.5 mM EDTA and 1% sodium lauryl sarcosine. Following completely washed away proteinase K with TE buffer, DNA in plugs were subsequently digested with RsaI and HinfI overnight at 37 °C. Plugs were loaded onto a 0.8% pulse-field agarose (Bio-Rad) gel in 0.5XTBE and run on a CHEF-DRII pulse field electrophoresis apparatus (Bio-Rad). The electrophoresis conditions were as follows: initial pulse 0.3 s, final pulse 16 s, voltage 6 V/cm, run time 13 h. The gels were dried and pre-hybridized in Church mix (0.5 M NaH$_2$PO$_4$, pH 7.2, 7% SDS), and then hybridized with telomeric repeat oligonucleotide probe γ-$^{32}$P-(CCCTAA)$_4$ in Church mix at 55 °C overnight. Gels were washed with 4XSSC, 0.1% SDS buffer at 55 °C and exposed to Phosphorimager screens overnight. The screen was scanned on a Typhoon Trio image system (GE Healthcare) and signals were analyzed with Imagequant TL (GE Life Sciences) software. Total activity in each lane was determined with ImageQuant software (Molecular Dynamics). For quantification of total telomeric DNA, the gels were deprobed with denature solution 0.5 N NaOH, 1.5 M NaCl and neutralized with 3 M NaCl, 0.5 M Tris-Cl, pH 7.0, and re-probed with telomeric probe γ-$^{32}$P-(CCCTAA)$_4$.

**TRAP assay**. TRAP assay was performed as described in the manufacture protocol (Millipore #S7700). Briefly, $1 \times 10^6$ cells were lysed with 200ul of CHAPS buffer and the soluble fractions were freshly frozen in liquid nitrogen and stored at −80 °C. The telomerase reaction was carried out in 25ul TRAP buffer containing: 20 mM Tris–Cl (pH 8.0), 63 mM KCl, 1.5 mM MgCl$_2$, 1 mM EGTA, 2 ng/μl of γ-$^{32}$P labeled TS primer (AATCCGTCGAGCAGAGTT), 50 μM 4dNTP and 1x TRAP primer mix at 30 °C for 40 min and quenched at 95 °C for 2 min. Following immediately adding Taq DNA polymerase, the PCR was performed as follows: 94 °C 30 s, 59 °C 30 s repeating for 28 times. The PCR reaction was separated on 10% acrylamide gel in 0.5XTBE (Invitrogen EC62752 1.0mmmx12w) and run for 81 min under 150 Voltage. The gel was dried and exposed to phosphorimager screen. The radioactive signals were captured and quantified as TRF Southern blotting.

**Homology modeling**. We generated homology structural models for the POT1a-TPP1 and POT1b-TPP1 using MODELLER[48] based on hPOT1$_{CTD}$-TPP1$_{PBM}$ structure (PDB: 5H65). All the structural figures are prepared by PyMOL Molecular Graphics System (Version 1.5 Schrödinger, LLC).

**Reporting summary**. Further information on research design is available in the Nature Research Reporting Summary linked to this article.

## Data availability

The data that support this study are available from the corresponding author upon reasonable request. Source data are provided with this paper.

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

## Acknowledgements
The authors would like to thank the members of the Chang lab for helpful suggestions. This work was supported by NIH RO1-CA202816 (S.C.), NIH RO3-252689 (S.C.), DOD W81XWH191005 (S.C.); NIH R01-AG050509 (J.N.), NIH R01-GM120094 (J.N.), American Cancer Society Research Scholar grant RSG-17-037-01-DMC (J.N.) and the Strategic Priority Research Program of the Chinese Academy of Sciences grant XDB37010303 (Y.C.).

## Author contributions
P.G. and S.C. conceived the project and designed the experiments. P.G., S.J. and T.T. performed biochemistry, cell biology and molecular biology experiments, V.M.T. and J.N. performed the telomerase extension assay and Y.C. conducted structural analysis. P.G. and S.C. analyzed and interpreted the data, composed the figures and wrote the paper.

## Competing interests
The authors declare no competing interests.
