## [Peer Review File · Nature Communications]

Distinct functions of POT1 proteins contribute to the regulation of telomerase recruitment to telomeresREVIEWER COMMENTS

Reviewer #1 (Remarks to the Author):

Chang: NatureComm ... POT1

A major question in the telomere field is how shelterin proteins function to protect chromosome ends from being recognized as damaged DNA while enabling telomerase-mediated telomere elongation. The key sequence-specific DNA-binding shelterin proteins are the duplex telomere binding proteins TRF1 and TRF2 and the telomere ssDNA binding protein POT1. A major mystery has been how human POT1 can both positively and negatively regulate telomere length. Here, in an elegant series of genetic studies, the authors exploit the experimental merits of the mouse and the fact that mice have two POT1 paralogs, enabling dissection of function and shedding important light on the mechanistic basis for POT1's dichotomy of function. Specifically, the authors elucidate independent roles of the two murine shelterin proteins POT1a and POT1b in telomere maintenance, and provide convincing evidence for POT1b's specific role in interacting with TPP1 to promote telomerase recruitment to telomeres and telomere elongation. The work convincingly demonstrates that loss of POT1b causes significant telomere attrition in a sarcoma model and MEFs, and reconstitution of POT1b rescues telomere attrition through its C-terminal TPP1 interacting domain, which occurs independent of the CST complex through additional depletion of CTC1. Lastly, a series of elegant and decisive biochemical and molecular experiments reveal that POT1b's three unique amino acids in the C-terminal HJRL domain is essential for its interaction with TPP1 to enhance telomerase recruitment to telomeres to promote telomere elongation. All the experiments are well performed and convincing, and the manuscript and figures are well-organized and clear. Below are several minor points whose clarification may strengthen this study.

It remains unclear why the authors utilized a sarcoma model heterozygous for Pot1a? The authors should explain their rationale for incorporating heterozygous loss of Pot1a?

Due to the observed significant telomere attrition and telomere free ends, it is surprising that no chromosomes exhibit non-reciprocal translocations in the provided images? Were any chromosomal events reminiscent of telomere induced crisis observed after Pot1b deletion in the p53-null model, such as non-reciprocal translocations or chromothripsis (micronuclei) observed?

In Fig. 3G and 3H, POT1bWT-expressing G2 mTert^{-/-}; Pot1b^{-/-} MEFs displayed shorter telomeres relative to GFP-expressing control group. What's the explanation for the reduction in telomere length in POT1bWT-expressing G2 mTert^{-/-}; Pot1b^{-/-} cells?

Fix typos: Supplemental Fig. 1F - Y axis title, Supplemental Fig. 2E – panel label.

Please check Myc-POT1a expression level in the immunoblot of Supplemental Fig. 9B and redo this control experiment.

Abstract: define CST for the general reviewer

Reference #1 was written in 1996 and should be replaced with a more recent general review.

Reviewer #2 (Remarks to the Author):

Shelterin protects telomeres from all aspects of the DNA damage response including ATM and ATR kinase signaling. The shelterin component POT1 forms a functional heterodimer with TPP1, binds to telomeric G-OH, prevents telomeres from being recognized as damaged DNA, and regulates telomerase activity at telomeres via TPP1.

Unlike most mammals including humans that have only one POT1 gene, the mouse processes two POT1 paralogs, POT1a and POT1b, which show distinct functions of human POT1 at telomeres, e.g., POT1a mainly functions to repress ATR-Chk1 DDR while POT1b prevent C strand resection and recruits CST for the C strand fill-in. In contrast to TPP1 that directly interacts with TERT and stimulates telomerase activity and processivity, human POT1 is showed to function both as a

negative and a positive regulator of telomerase activity at telomeres. This raises a question whether mouse POT1a and POT1b are distinguishable in terms of regulation of telomerase activity. In this work Gu et. al. establish Pot1aF/Δ; Pot1b-/-; p53 Δ/Δ and Pot1aF/Δ; Pot1b+/-; p53 Δ/Δ MEFs (abbreviated as Pot1b-/- and Pot1b+/-) and the derivative sarcoma cell lines. The authors demonstrate that: 1) reconstitution of Pot1b but not Pot1a in the Pot1b-/- cells promotes telomere elongation and rescue telomere-free chromosome fusions in a telomerase-dependent manner; 2) RAD51 (ALT pathway) and CST (C strand fill-in) are not required to POT1b-mediated telomere lengthening; 3) POT1b promotes TPP1 TEL patch-dependent telomeres recruitment; and 4) a unique cluster of POT1b amino acids, namely, D421, D426 and E428, are implicated in interacting with TPP1 to recruit telomerase to telomeres.

My overall appreciation is that this work will be of interest to a number of researchers in telomere and telomerase. This manuscript contains a substantial amount of good quality work, and experiments are well focused in supporting the role for POT1b in telomerase recruitment to telomeres. I think this work will be well suited for a high impact journal such as Nature Communications. I have only some minor points as listed below.

1. Supplemental Figure 2C-2E: The authors please indicate in the text or in the figure legends what the p53P/P (p53-R172P?) mutation is.
2. Supplemental Figure 5C: Please indicate the meaning of the numbers (0% and 72.5%) on the right side of the images.
3. Supplemental Figure 8A: Please indicate in the figure legend what antibodies are used for the blots.
4. Figure 6B and 6D: The interaction of POT1a and POT1b with TPP1 was compared in detail in Figure 6B. However, the POT1bLSAYKV (the desired POT1b construct bearing POT1a amino acids) loses that ability to interact with TPP1 (different from POT1a), suggesting that conversion is not successful as expected. So results of Figure 6E and 6F are not insufficient to reach the conclusion of "POT1b residues D421, D426, E428 in the HJRL domain.....".
5. The order of some panels needed to be rearranged according to their first reference in the text. For example, Supplemental Figure 7A and 7B, Supplemental Figure 8C and 8D, Figure 5A and Figure 4C.
6. "we generated a series of expression constructs where POT1aWT, POT1aF62A, POT1bWT and POT1bF62A are physically tethered to TPP1 with a 10 amino acid flexible linker (Figure 5A and Supplemental Figures 8A, 8B)": The schematic diagram in Figure 5A is confusing.
7. "we generated a series of POT1a/POT1b chimeras to ask whether the POT1b N-terminal OB folds (amino acids 1-350) or its C-terminus (aa 351-640) is required to mediate telomerase recruitment to telomeres (Figure 5A and Supplemental Figures 9A,9B)": The schematic diagram in Supplemental Figures 9A is sufficient. The reference to Figure 5A can be removed.
8. "Expressing either the POT1a1-350-POT1b351-640 chimeric protein (abbreviated POT1ab) or the POT1b1-350-POT1a351-640 (abbreviated POT1ba) in G3 Pot1b-/- sarcomas was unable to completely restore CST recruitment to telomeres (Supplemental Figure 9B)": The reference to Supplemental Figure 9C should be included here.
9. "However, only POT1ab was able to increase telomerase recruitment to telomeres by ~2-fold, similar to what we observed with POT1bWT expression (Figures 5B, 5C)": Remove the reference to Figure 5C.
10. "Using the Fucci system to examine the cell cycle profile of individual cells...": The reference to Supplemental Figure 9B can be included here.
11. Figure 1G and Figure 2F: It is confusing that POT1bWT expression reduces telomere-free fusions in Pot1b-/- MEFs (Figure 2F) but has no effect on that in Pot1b-/- sarcomas (Figure 1G).
12. "Since telomerase processivity appears identical in the presence of either POT1a-TPP1 or POT1b-TPP1, our results suggest that POT1b promotes telomerase recruitment to telomeres but is not required for telomerase processivity in vitro": Do the authors want to claim that 1) neither POT1a nor POT1b is required for telomerase processivity in vitro, or 2) both POT1a and POT1b promote telomerase processivity so POT1b is dispensable for telomerase processivity? Only Figure 4F is not sufficient for conclusion 1).
13. Figure 7A and 7C: Are data of Figures 7A and 7C from the same experiment? Data for POT1b and POT1b- TPP1ΔRD are identical in 7A and 7C while data for GFP and POT1b-TPP1 are different in 7A and 7C.
14. Figure 7D: I am wondering why POT1b-TPP1ΔRD (TPP1 lacking amino acids 159 to 246 that interacts with POT1 proteins) is unable to interact with endogenous TPP1 in Pot1b-/- cells to restore telomere length to a certain extent. Could it be because the flexible linker is short?

15. Figure 5F: The authors please define in the figure legend what “Pot1 positive cells” are.

16. Figure 5E and 5F: It is intriguing that POT1b (but not POT1a) exhibits cell cycle-regulated telomere localization and accumulates at telomeres during S/G2, which is consistent with its role in promoting telomerase recruitment. Can the authors briefly speculate the underlying mechanism for cell cycle preference of POT1b's telomere localization.

17. “Our discovery that POT1a and POT1b possess distinct telomere length regulatory functions reveal how human POT1 functions both as a negative and a positive regulator of telomerase activity at telomeres”: The reasoning is not so convincing. POT1a and POT1b are two different proteins, which possess distinct characteristics in interaction with TPP1, telomere localization and telomerase recruitment. One can easily imagine that these two paralogs compete with each other for access to the 3' G-overhang, so that they positively and negatively regulate telomerase activity, respectively. However, there is only one POT1 in human and all hPOT1 are identical in binding with TPP1 and 3' G-overhang. The difference between POT1a and POT1b unveiled in this manuscript is insufficient to reveal how human POT1 functions both as a negative and a positive regulator of telomerase activity at telomeres. By the way, hPOT1-TPP1 as well as mPOT1b-TPP1, interacts with the CST complex that is supposed to limit telomerase activity at telomeres in human and in yeast. The interactions between POT1-TPP1 and telomerase and the CST complex may also render POT1-TPP1 as a negative and a positive regulator of telomerase activity at telomeres.

Reviewer #1 (Remarks to the Author):

We thank reviewer 1 for the very encouraging comments regarding this manuscript.

1. It remains unclear why the authors utilized a sarcoma model heterozygous for Pot1a? The authors should explain their rationale for incorporating heterozygous loss of Pot1a?

From our biochemical analysis, we knew that POT1a competes with POT1b for access to the 3' G-overhang. We therefore reasoned that depleting (not deleting) POT1a will enable us to detect any subtle telomere length related phenotypes exhibited by POT1b, while still having enough POT1a to provide genome stability. For our analysis of the telomere maintenance functions of POT1b, we generated both *Pot1a^{F/A}; Pot1b^{-/-}; p53^{F/F}* and *Pot1a^{F/F}; Pot1b^{-/-}; p53^{F/F}* MEFs. Ad-Cre administration resulted in the deletion of the floxed *Pot1a* alleles. We discovered that *Pot1a^{Δ/Δ}; Pot1b^{-/-}; p53^{Δ/Δ}* MEFs are highly genomically unstable, generating many chromosomal aberrations (Reviewer Figure 1). This increased instability is likely to inhibit tumor formation. In contrast, *Pot1a^{F/A}; Pot1b^{-/-}; p53^{Δ/Δ}* MEFs are relatively genomically stable with ~18% fusions (Figures 1C, 1G), no TIFs and readily formed sarcomas when injected into the flanks of SCID mice. Because POT1a inhibits telomerase access to telomeres, we were able to easily detect telomere elongation by POT1b^{WT} in *Pot1a^{F/A}; Pot1b^{-/-}; p53^{Δ/Δ}* MEFs where POT1a levels are low.

2. Due to the observed significant telomere attrition and telomere free ends, it is surprising that no chromosomes exhibit non-reciprocal translocations in the provided images? Were any chromosomal events reminiscent of telomere induced crisis observed after Pot1b deletion in the p53-null model, such as non-reciprocal translocations or chromothripsis (micronuclei) observed?

Reviewer Figure 1: Increased chromosome aberrations in immortalized G2; *Tert*^{-/-}; *Pot1b*^{-/-} MEFs. A. Primary MEFs harvested at day 2. B. SV40-immortalized MEF harvested at day 30. C. SV-40 immortalized MEF harvested at day 50. D. Quantification of percentage of fusions per chromosome. White arrows: fusions without telomeres at fusion sites; red arrows: fusions with telomeres at fusion sites. C: circularized chromosomes.

In SV-40 immortalized G2 *mTerc*^{-/-}; *Pot1b*^{-/-} MEFs passaged for 50 days, we observed chromosome fusions reminiscent of those induced by critically shortened telomeres during crisis (Reviewer Figure 1). We detected trains of multiple end-to-end chromosome fusions without telomeres at fusion sites and even circularized chromosomes without any telomeric signals at fusion sites. Interestingly, we did not detect any micronuclei in these cells. *Pot1a^{Δ/Δ}; Pot1b^{-/-}; p53^{Δ/Δ}* MEFs also display increased end-to-end chromosome fusions, with fusions observed in 100% of metaphases examined (Reviewer Figure 2). In contrast, *Pot1a^{Δ/Δ}; Pot1b^{+/-}; p53^{Δ/Δ}* MEFs displayed ~10-fold less fused chromosome, suggesting that POT1b plays a role in telomere end protection in the absence of functional POT1a.

3. In Fig. 3G and 3H, POT1b^{WT}-expressing G2 *mTert*^{-/-}; *Pot1b*^{-/-} MEFs displayed shorter telomeres relative to GFP-expressing control group. What's the explanation for the reduction in telomere length in POT1b^{WT}-expressing G2 *mTert*^{-/-}; *Pot1b*^{-/-} cells?

The reviewer raise a very interesting question. Reconstitution with POT1b in G2 *mTert*^{-/-}; *Pot1b*^{-/-} MEFs promotes the recruitment of CST to telomere to efficiently fill-in the 5' C-strand, eliminating the G-overhang which would be detected by Q-FISH as a reduction in telomere signal. However, in the GFP control, the longer G-overhang would result in increased telomere signals from the Q-FISH analysis.

4. Fix typos: Supplemental Fig. 1F - Y axis title, Supplemental Fig. 2E – panel label.

Fixed

Reviewer Figure 2. Chromosome aberration in parental $P53^{\Delta/\Delta}$; $Pot1a^{F/\Delta}$; $Pot1b^{+/-}$ (#1, #4, #9) and parental $P53^{\Delta/\Delta}$; $Pot1a^{F/\Delta}$; $Pot1b^{-/-}$ (#5) MEFs. A. examples of CO-FISH images. TTAGGG (red), CCCTAA (green). **B.** percentage of fusions per metaphase in A. **C.** percentage of metaphase with telomere end-to-end fusions in A. **D.** Quantification of chromosome aberrations. Yellow arrow: fusions without telomere signal at sites of fusion; red arrows: fusions with telomere signal at sites of fusion; green arrow: fusions of multiple chromosomes.

5. Please check Myc-POT1a expression level in the immunoblot of Supplemental Fig. 9B and redo this control experiment.

We have redone this Western blot and placed it in new Supplemental Figure 9B (Reviewer figure 3).

6. Abstract: define CST for the general reviewer

Done

7. Reference #1 was written in 1996 and should be replaced with a more recent general review.

We replaced it with a newer reference.

Reviewer Figure 3. Supplemental figure 9B. Cells expressing the indicated epitope-tagged DNA constructs were lysed and analyzed by Western analysis. Anti-Myc and anti-Flag antibodies were mixed together to detect expressed POT1 proteins. γ -tubulin was used as loading control.

Reviewer #2 (Remarks to the Author):

We thank the reviewer for stating that “this work will be of interest to a number of researchers in telomere and telomerase. This manuscript contains a substantial amount of good quality work, and experiments are well focused in supporting the role for POT1b in telomerase recruitment to telomeres. I think this work will be well suited for a high impact journal such as Nature Communications.”

1. Supplemental Figure 2C-2E: The authors please indicate in the text or in the figure legends what the p53^{P/P} (p53-R172P?) mutation is.

“p53^{P/P} is the homozygous p53 R172P mutation. We have inserted this into the figure legend.

2. Supplemental Figure 5C: Please indicate the meaning of the numbers (0% and 72.5%) on the right side of the images. The number at the right side indicate the percentage of nuclei with more than 5 Pot1b foci colocalized with telomeres. We inserted this into the figure legend.

3. Supplemental Figure 8A: Please indicate in the figure legend what antibodies are used for the blots.

We mixed antibodies against epitope tags (anti-Flag, anti-HA and anti-Myc) together in the Western blot analysis. We inserted this in the figure legend in supplemental figure 8B.

4. Figure 6B and 6D: The interaction of POT1a and POT1b with TPP1 was compared in detail in Figure 6B. However, the POT1bLSAYKV (the desired POT1b construct bearing POT1a amino acids) loses that ability to interact with TPP1 (different from POT1a), suggesting that conversion is not successful as expected. So results of Figure 6E and 6F are not insufficient to reach the conclusion of “POT1b residues D421, D426, E428 in the HJRL domain.....”.

Mutating POT1b residues D421, D426 and E428 into POT1a residues A421, Y426 and K428 revealed that POT1b^{AYK} cannot interact with TPP1 and therefore cannot recruit telomerase to telomeres. These results suggest that the POT1b DDE residues are required for POT1b to interact with TPP1. Since POT1b^{AYK} does not localize to telomeres, we tethered it to TPP1^{WT} to enable localization to telomeres (Figures 7A, 7B). However, POT1b^{AYK}-TPP1^{WT} still does not elongate telomeres, indicating that POT1b's DDE must contact TPP1 to promote telomerase recruitment. We have modified the manuscript to make this point clearer.

5. The order of some panels needed to be rearranged according to their first reference in the text. For example, Supplemental Figure 7A and 7B, Supplemental Figure 8C and 8D, Figure 5A and Figure 4C.

We agree and have made the suggested changes.

6. “we generated a series of expression constructs where POT1a^{WT}, POT1a^{F62A}, POT1b^{WT} and POT1b^{F62A} are physically tethered to TPP1 with a 10 amino acid flexible linker (Figure 5A and Supplemental Figures 8A, 8B)”: The schematic diagram in Figure 5A is confusing.

We agree and have made the required changes. We remove the tethered construct figures in Figure 5A into Supplemental Figure 8A and moved original Supplemental Figure 9A into Figure 5A and included the corresponding changes in the text and figure legends.

7. “we generated a series of POT1a/POT1b chimeras to ask whether the POT1b N-terminal OB folds (amino acids 1-350) or its C-terminus (aa 351-640) is required to mediate telomerase recruitment to telomeres (Figure 5A and Supplemental Figures 9A,9B)”: The schematic diagram in Supplemental Figures 9A is sufficient. The reference to Figure 5A can be removed.

We agree with this point and deleted original Supplemental Figure 9A and kept it as new Figure 5A

8. “Expressing either the POT1a1-350-POT1b351-640 chimeric protein (abbreviated POT1ab) or the POT1b1-350-POT1a351-640 (abbreviated POT1ba) in G3 Pot1b^{-/-} sarcomas was unable to completely restore CST recruitment to telomeres (Supplemental Figure 9B)”: The reference to Supplemental Figure 9C should be included here.

We added the reference to Supplemental Figure 9B in the text.

9. “However, only POT1ab was able to increase telomerase recruitment to telomeres by ~2-fold, similar to what we observed with POT1b^{WT} expression (Figures 5B, 5C)”: Remove the reference to Figure 5C.

We removed this reference.

10. “Using the Fucci system to examine the cell cycle profile of individual cells...”: The reference to Supplemental Figure 9B can be included here.

We add the reference to Supplemental Figure 9C in the text.

11. Figure 1G and Figure 2F: It is confusing that POT1b^{WT} expression reduces telomere-free fusions in Pot1b^{-/-} MEFs (Figure 2F) but has no effect on that in Pot1b^{-/-} sarcomas (Figure 1G).

In Figure 2F, the genotype of the MEF used was *CAG-Cre^{ER}; Pot1b^{-/-}; CTC1^{F/F}* while the genotype of the sarcoma in Figure 1G was *Pot1b^{-/-}*. In the former, 4-HT treatment results in very rapid telomere shortening and formation of new fused chromosomes. Expression of POT1b^{WT} reduced the number of these new fusions. In contrast, *Pot1b^{-/-}* sarcomas shorten telomeres much more slowly, and new fusions were not generated. Thus the expression of POT1b^{WT} does not significantly reduce the number of chromosome fusions observed (the fusions formed earlier still persist).

12. “Since telomerase processivity appears identical in the presence of either POT1a-TPP1 or POT1b-TPP1, our results suggest that POT1b promotes telomerase recruitment to telomeres but is not required for telomerase processivity in vitro”:

Do the authors want to claim that 1) neither POT1a nor POT1b is required for telomerase processivity *in vitro*, or 2) both POT1a and POT1b promote telomerase processivity so POT1b is dispensable for telomerase processivity? Only Figure 4F is not sufficient for conclusion 1).

We believe that both POT1a and POT1b can promote the telomerase processivity when analyzed by the *in vitro* telomerase processivity assay. In the direct telomerase processivity assay used in Figure 4F, the artificial G-overhangs are very short and POT1a/b-TPP1 are present in excess. Under this condition, the binding of either POT1a or POT1b to the 3' overhang enables TPP1 to recruit telomerase to the G-overhang to promote telomerase processivity. However, in our *in vivo* telomerase recruitment assay, we monitor the telomerase loading on telomeres under physiological conditions, where TPP1 is in the complex of TRF1/2-TIN2-TPP1-POT1 on telomeres. In this situation, the requirement for TPP1 to recruit telomerase to telomere requires POT1b and not POT1a.

13. Figure 7A and 7C: Are data of Figures 7A and 7C from the same experiment? Data for POT1b and POT1b- TPP1 Δ RD are identical in 7A and 7C while data for GFP and POT1b-TPP1 are different in 7A and 7C.

Yes, it was from the same experiment. We felt it was clearer to split our results into two figures.

14. Figure 7D: I am wondering why POT1b-TPP1 Δ RD (TPP1 lacking amino acids 159 to 246 that interacts with POT1 proteins) is unable to interact with endogenous TPP1 in Pot1b $^{-/-}$ cells to restore telomere length to a certain extent. Could it be because the flexible linker is short?

TPP1 Δ RD cannot interact with POT1b and tethering it to POT1b does not restore telomerase recruitment. This suggests that physical interaction between POT1b and TPP1 is required for telomerase recruitment. In addition, since the linker is only 10 amino acids, POT1b tethered to TPP1 Δ RD cannot interact with endogenous TPP1 to recruit telomerase. Finally, TPP1 Δ RD functions as a dominant negative in the presence of WT TPP1, also preventing tethered POT1b from interacting with endogenous TPP1.

15. Figure 5F: The authors please define in the figure legend what "Pot1 positive cells" are.

Pot1 positive cells mean wildtype MEF. Changed from "The distribution of Pot1 positive cells in different cell cycle..." to "The distribution of POT1a or POT1b in wildtype MEF during different cell cycle..." in the text of Materials and Methods section. Also changed "WT MEF" in figure legend Fig 5F and supplemental fig 9D to "wildtype"

16. Figure 5E and 5F: It is intriguing that POT1b (but not POT1a) exhibits cell cycle-regulated telomere localization and accumulates at telomeres during S/G2, which is consistent with its role in promoting telomerase recruitment. Can the authors briefly speculate the underlying mechanism for cell cycle preference of POT1b's telomere localization.

This is a fascinating question and we do not know the mechanism to explain this. We are exploring the possibility that POT1a and POT1b are cell cycle regulated through post-translational modifications.

17. "Our discovery that POT1a and POT1b possess distinct telomere length regulatory functions reveal how human POT1 functions both as a negative and a positive regulator of telomerase activity at telomeres": The reasoning is not so convincing. POT1a and POT1b are two different proteins, which possess distinct characteristics in interaction with TPP1, telomere localization and telomerase recruitment. One can easily imagine that these two paralogs compete with each other for access to the 3' G-overhang, so that they positively and negatively regulate telomerase activity, respectively. However, there is only one POT1 in human and all hPOT1 are identical in binding with TPP1 and 3' G-overhang. The difference between POT1a and POT1b unveiled in this manuscript is insufficient to reveal how human POT1 functions both as a negative and a positive regulator of telomerase activity at telomeres. By the way, hPOT1-TPP1 as well as mPOT1b-TPP1, interacts with the CST complex that is supposed to limit telomerase activity at telomeres in human and in yeast. The interactions between POT1-TPP1 and telomerase and the CST complex may also render POT1-TPP1 as a negative and a positive regulator of telomerase activity at telomeres.

We agree with the sentiments of this reviewer and have modified the abstract to reflect this point. There is only one hPOT1 protein encoded by the human genome. We reason that hPOT1 must possess the functions observed in both POT1a and POT1b proteins. Based on our *in vitro* and *in vivo* studies, removal of hPOT1 or introduction of hPOT1 OB1/2 mutations lead to increased TIFs and elongation of telomere length. These phenotypes are functionally similar to what we observed when POT1a is deleted. However, hPOT1 is required for telomerase processivity *in vitro*, which means hPOT1 also plays a role in the telomerase recruitment and maintenance of telomere elongation. We are currently exploring which amino acid residues in hPOT1 are required for telomerase recruitment.

REVIEWERS' COMMENTS

Reviewer #1 (Remarks to the Author):

Strong paper ... recommend publication

Reviewer #2 (Remarks to the Author):

I have read through the rebuttal comments and the revised manuscript. The authors have satisfactorily answered and addressed my concerns, and I therefore now think that the manuscript is acceptable for publication.